# BLAST: Block-Level Adaptive Structured Matrices for Efficient Deep Neural Network Inference

**Changwoo Lee    Soo Min Kwon    Qing Qu    Hun-Seok Kim**
University of Michigan
{cwoolee,kwonsm,qingqu,hunseok}@umich.edu

## Abstract

Large-scale foundation models have demonstrated exceptional performance in language and vision tasks. However, the numerous dense matrix-vector operations involved in these large networks pose significant computational challenges during inference. To address these challenges, we introduce the Block-Level Adaptive STructured (BLAST) matrix, designed to learn and leverage efficient structures prevalent in the weight matrices of linear layers within deep learning models. Compared to existing structured matrices, the BLAST matrix offers substantial flexibility, as it can represent various types of structures that are either learned from data or computed from pre-existing weight matrices. We demonstrate the efficiency of using the BLAST matrix for compressing both language and vision tasks, showing that (i) for medium-sized models such as ViT and GPT-2, training with BLAST weights boosts performance while reducing complexity by 70% and 40%, respectively; and (ii) for large foundation models such as Llama-7B and DiT-XL, the BLAST matrix achieves a 2x compression while exhibiting the lowest performance degradation among all tested structured matrices. Our code is available at https://github.com/changwoolee/BLAST.

## 1 Introduction

Foundation models built on large deep neural networks (DNNs) have demonstrated remarkable performance in vision and language tasks. However, the size of these large networks poses both computational and storage challenges, especially in resource-constrained environments such as edge devices. The size of a single DNN often exceeds the capacity of the supporting hardware devices [1–5]. For example, Llama-70B [1] demands at least 140GB of memory solely for loading its weights in half-precision floating point representation, while the state-of-the-art commercial GPU only accommodates 80GB of memory. Furthermore, inference with these networks involves numerous dense matrix-vector operations, which can be limiting when computing power is constrained.

Fortunately, large (overparameterized) DNNs often exhibits parameter redundancy, where the intrinsic dimension of the weights is much lower than the ambient dimension. As such, the weights should be *structured*, possessing hidden properties such as low-rankness [6–9] or sparsity [10, 11]. Hence, it is possible to replace (or factorize) these dense existing weight matrices with structured ones without degrading performance [10–12]. However, using structured matrices that do not align with the true underlying structure of the weight matrices can result in significant performance degradation. We demonstrate this point in Figure 1 where we attempt to capture the structure of a diffusion model transformer (DiT) [13] using the low-rank structure to generate synthetic images. In Figure 1, we compress the model's linear layers by approximately 50% of the total number of parameters using low-rank weight matrices via singular value decomposition (SVD) and generate images with the compressed model (see Section 4.2 and Appendix C.3 for details). As shown in Figure 1 (middle), simply using the low-rank structure introduces unwanted artifacts in the generated images.

38th Conference on Neural Information Processing Systems (NeurIPS 2024).

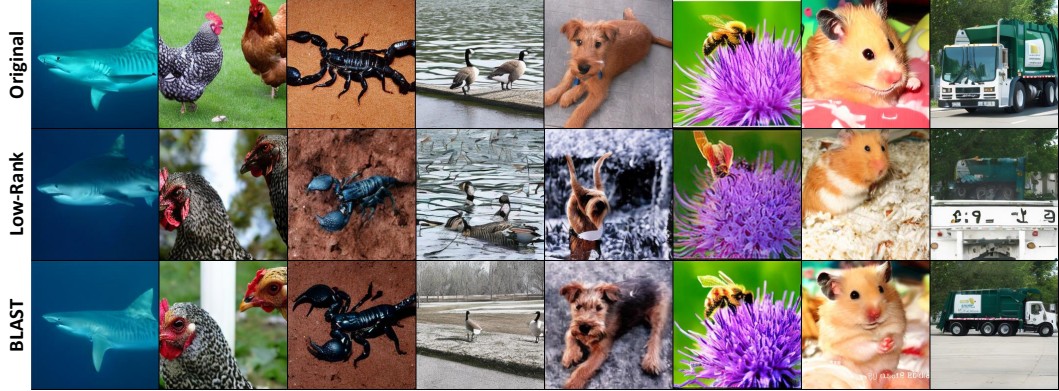

Figure 1: Examples of generated images using DiT [13] starting from the same noise vectors and a deterministic solver. The original model is compressed by 50% through BLAST or low-rank matrices and re-trained for 10 epochs on ImageNet. The images from the model compressed via BLAST preserve the quality of the images of the original model, whereas the images generated by the low-rank model contain more undesired artifacts.

To address this issue, many flexible structures for modeling DNN weights have been proposed to minimize the misalignment between imposed and true low-dimensional structures. For example, Dao et al. [14] proposed the Monarch matrix, a specific type of Block Low-Rank (BLR) structure [15], in which all blocks share the same rank, intended for use in the linear layers of transformers [16]. Matrix multiplication with a Monarch matrix can be performed efficiently using batched matrix multiplication routines. Additionally, Chen et al. [17] investigated a block sparse plus low-rank structure. However, all of these methods still suffer from the fact that the underlying structure of each weight matrix is not known a priori. By imposing one of these structures, performance degradation may still occur due to misalignment. Recently, Lee and Kim [12] introduced a data-driven design called Generalized Block Low-Rank (GBLR). This approach employs multiple rank-1 blocks with various sizes and locations learned from data via differentiable masks. Unfortunately, the GBLR matrix is optimized for custom-designed hardware, as the learned block patterns are random. It has limited usability on general GPUs as the computation of GBLR matrices does not accelerate well on them.

In this work, we introduce the Block-Level Adaptive Structured (BLAST) matrix, a versatile and efficient design tailored to uncover various low-dimensional structures in the weight matrices of DNNs for accelerated inference on GPUs. Our matrix structure leverages shared bases across block matrices with block-wise diagonal coupling factors. This structure encapsulates different structures such as low-rank, block low-rank, block-diagonal matrices, and their combinations. BLAST matrices can be applied to the training scenario from scratch or compression after training. For training from scratch, we let the linear layers of the DNN to directly adopt the BLAST structure and learn its factors from data. The factors of the BLAST matrix are constructed to have well-defined gradients, allowing them to be optimized using popular methods like stochastic gradient descent (SGD) or Adam [18]. For compressing existing weights, we propose a factorization algorithm to learn the BLAST factors from pre-trained weights. The compression performance can be further improved by updating the BLAST factors using data, a process we call "re-training".

We demonstrate the efficiency of BLAST by training Vision Transformers (ViT) [19] and GPT-2 [20] from scratch on various datasets, showing that it can reduce complexity by 70% and 40%, respectively. We also compress existing ViT and Diffusion Transformer (DiT) [13] models with BLAST matrices by 70% and 50%, respectively, demonstrating that BLAST compression (and re-training) achieves higher accuracy / quality compared to existing methods for ViT and DiT (see Figure 1). For the language tasks, we compress Llama-7B [1] by 50% via BLAST and re-train on 0.49B tokens, showing the lowest accuracy degradation with significant inference speedup on a NVIDIA A100 GPU. Overall, our contributions can be summarized as follows:

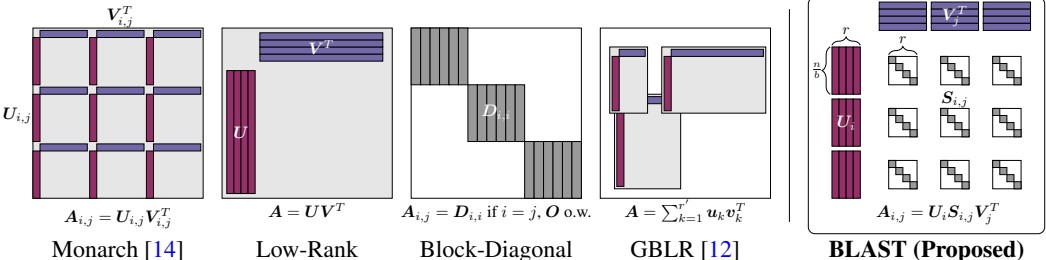

Figure 2: Existing structured matrices and our proposed BLAST matrix. The unique structure of BLAST allows for flexible matrix structures while enabling faster matrix multiplication compared to existing matrices.

- We propose a novel block-structured matrix called BLAST that encompasses a wide range of matrix structures, allowing for *faster matrix multiplication*. Various existing structured matrices such as Low-Rank, Monarch [14], and Block Diagonal matrices can be expressed using the BLAST matrix.

- We provide gradient descent-based methods to find the BLAST factors for DNN weights. We empirically show that standard DNN training with the BLAST weight matrices effectively recovers the original accuracy while achieving up to a 70% reduction in computational complexity.

- In cases where pre-trained dense weights are available, we propose a preconditioned gradient descent factorization algorithm to decompose the weights to BLAST factors for compression and further re-training. Our experimental results show that pre-trained foundation models for vision or language tasks can be compressed by 50% using BLAST matrices.

**Notation and Organization.** We use $\sigma_1(\boldsymbol{X})$ to denote the largest singular value of the matrix $\boldsymbol{X}$. The notation $\odot$ indicates Hadamard product.

The rest of the paper is organized as follows. In Section 2, we introduce the BLAST matrix and discuss its properties. In Section 3, we propose a methodology to train/compress DNNs with BLAST weight matrices. In Section 4, we demonstrate the effectiveness of the BLAST weights in improving efficiency without noticeable accuracy degradation. We discuss related works in Section 5, and conclude in Section 6.

## 2 Block-Level Adaptive Structured (BLAST) Matrix

Consider a square matrix[1] $\boldsymbol{A} \in \mathbb{R}^{n \times n}$ for some $n \in \mathbb{N}$, which has an unknown intrinsic low-dimensional structure. We first equally partition the matrix $\boldsymbol{A}$ into $b \times b$ blocks of size $p \times p$ where $b, p \in \mathbb{N}$ are constants such that $n = bp$:

$$\boldsymbol{A} = \begin{bmatrix} \boldsymbol{A}_{1,1} & \boldsymbol{A}_{1,2} & \cdots & \boldsymbol{A}_{1,b} \\ \boldsymbol{A}_{2,1} & \boldsymbol{A}_{2,2} & \cdots & \boldsymbol{A}_{2,b} \\ \vdots & \vdots & \ddots & \vdots \\ \boldsymbol{A}_{b,1} & \boldsymbol{A}_{b,2} & \cdots & \boldsymbol{A}_{b,b} \end{bmatrix}, \quad \boldsymbol{A}_{i,j} \in \mathbb{R}^{p \times p}, \quad i, j \in [b]. \tag{1}$$

Then, the BLAST matrix parameterizes each block matrix $\boldsymbol{A}_{i,j}$ using three factors:

$$\boldsymbol{A}_{i,j} = \boldsymbol{U}_i \boldsymbol{S}_{i,j} \boldsymbol{V}_j^T, \tag{2}$$

where $\boldsymbol{U}_i, \boldsymbol{V}_j \in \mathbb{R}^{p \times r}$ are the left and right factors, respectively, and $\boldsymbol{S}_{i,j} = \mathrm{diag}(\boldsymbol{s}_{i,j})$ is an $r \times r$ diagonal matrix whose diagonal entries are $\boldsymbol{s}_{i,j} \in \mathbb{R}^r$. We provide a visual representation on the rightmost side of Figure 2, and illustrate how this structure differs from other types of matrices. While the BLAST structure may appear similar to SVD, there are two notable differences: (i) the left and right factors do *not* need to be orthonormal, and (ii) the diagonal entries do *not* need to be positive. These distinctions make it more flexible in capturing different types of low-rank structures.

As illustrated in Figure 2, the BLAST matrix also comes with two unique properties:

---

[1] For an $m \times n$ rectangular matrix, we partition $m$ rows into $b$ chunks assuming that $b$ divides $m$ as well.

- **Factor Sharing:** The left factor matrix $\boldsymbol{U}_i$ of size $rp$ is *shared* across $b$ blocks at the $i^{\text{th}}$ row, i.e., $\boldsymbol{A}_{i,1}, \ldots, \boldsymbol{A}_{i,b}$. Likewise, the right factor $\boldsymbol{V}_j$ is shared across the blocks at the $j^{\text{th}}$ column. On the other hand, the diagonal factor $\boldsymbol{s}_{i,j}$ of size $r$ is specific to each block $\boldsymbol{A}_{i,j}$. Hence the total number of parameters of an $n \times n$ BLAST matrix with $b \times b$ number of blocks of rank $r$ is $2rpb + rb^2 = 2nr + rb^2$. This reduces the number of parameters $b$ times by enforcing the blocks at the same row or column share the same bases.

- **Individual Diagonal Factors:** The individual diagonal factors of each block matrix are the source of the adaptivity and flexibility of the BLAST matrix. By changing the values of the diagonal factors, the BLAST matrix can encompass a wide variety of matrix structures. These factors can be estimated using *gradient descent*, since $\boldsymbol{s}_{i,j}$ is a real-valued vector and $\boldsymbol{A}_{i,j} = \boldsymbol{U}_i \text{diag}(\boldsymbol{s}_{i,j}) \boldsymbol{V}_j^T$ is linear to $\boldsymbol{s}_{i,j}$.

**Low-Rank Matrices as Special Cases of BLAST** To demonstrate how the BLAST matrix can capture different types of structures, we present an example showing how the BLAST matrix can encompass a low-rank matrix. Consider the case where all the diagonal factors are ones, i.e., $\boldsymbol{s}_{i,j} = \mathbf{1}_r$ for all $i, j = 1, 2, \ldots, b$. Then, we can write the block matrix as follows:

$$\boldsymbol{U}\boldsymbol{V}^T = \begin{bmatrix} \boldsymbol{U}_1 \\ \boldsymbol{U}_2 \\ \vdots \\ \boldsymbol{U}_b \end{bmatrix} \begin{bmatrix} \boldsymbol{V}_1 & \boldsymbol{V}_2 & \cdots \boldsymbol{V}_b \end{bmatrix} = \begin{bmatrix} \boldsymbol{U}_1\boldsymbol{V}_1^T & \boldsymbol{U}_1\boldsymbol{V}_2^T & \cdots & \boldsymbol{U}_1\boldsymbol{V}_b^T \\ \boldsymbol{U}_2\boldsymbol{V}_1^T & \boldsymbol{U}_2\boldsymbol{V}_2^T & \cdots & \boldsymbol{U}_2\boldsymbol{V}_b^T \\ \vdots & \vdots & \ddots & \vdots \\ \boldsymbol{U}_b\boldsymbol{V}_1^T & \boldsymbol{U}_b\boldsymbol{V}_2^T & \cdots & \boldsymbol{U}_b\boldsymbol{V}_b^T \end{bmatrix}.$$

Hence, if the true underlying structure is low-rank, we can expect the BLAST matrix to learn this specific structure. Similarly, we show in Section A.1 that the BLAST matrix can construct *low-rank*, *block-diagonal*, and *block low-rank* matrices through different diagonal parameters. A combination of these canonical structured matrices, such as a *low-rank with block-diagonal* matrix, can also be achieved by simply concatenating the factors of each matrix.

**Matrix Multiplication** DNNs involve numerous matrix-vector (matrix-matrix) multiplications in the form of $\boldsymbol{y} = \boldsymbol{A}\boldsymbol{x}$ ($\boldsymbol{Y} = \boldsymbol{A}\boldsymbol{X}$). Algorithm 1 depicts the BLAST matrix-vector multiplication procedure. Consider the partitioned input vector $\boldsymbol{x} = [\boldsymbol{x}_1^T, \boldsymbol{x}_2^T, \cdots, \boldsymbol{x}_b^T]^T$ and the partitioned output vector $\boldsymbol{y} = [\boldsymbol{y}_1^T, \boldsymbol{y}_2^T, \cdots, \boldsymbol{y}_b^T]^T$. The $i^{\text{th}}$ partitioned output vector $\boldsymbol{y}_i$ is then computed by the sum of the $b$ block-wise matrix-vector multiplications along $j = 1, \ldots, b$:

---

**Algorithm 1** BLAST Matrix-Vector Product

**Require:** $\boldsymbol{U}, \boldsymbol{V}, \boldsymbol{s}, \boldsymbol{x}$
1: $[\boldsymbol{x}_1^T, \boldsymbol{x}_2^T, \cdots, \boldsymbol{x}_b^T]^T \leftarrow \boldsymbol{x}$
2: **for** $j = 1, 2, \ldots, b$ **do** $\triangleright$ #Parallel
3: $\quad \boldsymbol{z}_j \leftarrow \boldsymbol{V}_j^T \boldsymbol{x}_j$
4: **end for**
5: **for** $i = 1, 2, \ldots, b$ **do** $\triangleright$ #Parallel
6: $\quad \boldsymbol{y}_i \leftarrow \boldsymbol{U}_i \sum_{j=1}^b \boldsymbol{s}_{i,j} \odot \boldsymbol{z}_j$
7: **end for**
8: **return** $\boldsymbol{y} \leftarrow [\boldsymbol{y}_1^T, \ldots, \boldsymbol{y}_b^T]^T$

---

$$\boldsymbol{y}_i = \sum_{j=1}^b \boldsymbol{A}_{i,j}\boldsymbol{x}_j = \sum_{j=1}^b \boldsymbol{U}_i \boldsymbol{S}_{i,j} \boldsymbol{V}_j^T \boldsymbol{x}_j = \boldsymbol{U}_i \left( \sum_{j=1}^b \boldsymbol{S}_{i,j} \left( \boldsymbol{V}_j^T \boldsymbol{x}_j \right) \right), \quad i = 1, \ldots, b. \tag{3}$$

The number of multiplications required to perform the matrix-vector multiplication $\boldsymbol{y} = \boldsymbol{A}\boldsymbol{x}$ is $(2n + b^2)r$. The matrix multiplication $\boldsymbol{z}_j = \boldsymbol{V}_j^T \boldsymbol{x}_j$, $j = 1, \ldots, b$ is computed once and shared across $i = 1, \ldots, b$, whereas the matrix multiplications in Line 3 and Line 6 of Algorithm 1 can be executed in parallel, e.g., by `torch.bmm` in PyTorch [21]. An implementation of Algorithm 1 for general matrix or tensor inputs can be found in Appendix A.

## 3 Applications of BLAST Matrices

There are two main applications of BLAST matrices: (i) *training from scratch* with the BLAST structure and (ii) *compression of pre-trained weights* using BLAST factorization.

### 3.1 Training from Scratch using BLAST Matrices

To train a DNN on a dataset, parameters are typically initialized randomly and updated through stochastic gradient descent. In this setting, BLAST can replace dense weights to learn structures

from the training data. Instead of using random dense weight matrices, the model is initialized with random BLAST factors $U_i, V_j, s_{i,j}$. Since the forward and the backward path of the linear layer involving the weight matrix is composed of three linear operations as in Equation (3), the derivatives of the minibatch loss can be back-propagated by automatic differentiation frameworks [21]. Hence, all of the trainable parameters of BLAST can be updated using conventional optimizers (e.g., Adam [18] or AdamW [22]) without additional treatment.

## 3.2 Compressing Weights via BLAST Factorization

**BLAST Factorization via Gradient Descent**    Given pre-trained dense weights of a DNN, we can compress the weights using BLAST matrices. Let $A$ denote the weight matrix and $A_{i,j}$ denote its blocks. We estimate the BLAST factors of $A_{i,j}$ by finding the factors of the BLAST matrix that minimize the Frobenius norm error between the original weight matrix and the BLAST structure:

$$\ell(U_*, V_*, s_{*,*}) = \sum_{i=1}^{b}\sum_{j=1}^{b} \frac{1}{2} \left\| A_{i,j} - U_i \text{diag}(s_{i,j}) V_j^T \right\|_F^2, \tag{4}$$

where $*$ denotes the collection of all $b$ components along the axis. This problem shares many characteristics with the classical matrix factorization problem [23–26], and hence we can solve for the factors using alternating gradient descent starting from small random initialization (e.g., Line 1 of Algorithm 2) [27, 8]. That is, the $k^{\text{th}}$ gradient descent step is composed of three alternating updates with a step size $\eta > 0$:

$$U_i^{(k+1)} \leftarrow U_i^{(k)} - \eta_{U_i^{(k)}} \cdot \nabla_{U_i^{(k)}} \ell\left(U_*^{(k)}, V_*^{(k)}, s_{*,*}^{(k)}\right), \tag{5}$$

$$V_j^{(k+1)} \leftarrow V_j^{(k)} - \eta_{V_j^{(k)}} \cdot \nabla_{V_j^{(k)}} \ell\left(U_*^{(k+1)}, V_*^{(k)}, s_{*,*}^{(k)}\right), \tag{6}$$

$$s_{i,j}^{(k+1)} \leftarrow s_{i,j}^{(k)} - \eta_{s_{i,j}^{(k)}} \cdot \nabla_{s_{i,j}^{(k)}} \ell\left(U_*^{(k+1)}, V_*^{(k+1)}, s_{*,*}^{(k)}\right). \tag{7}$$

With properly chosen step sizes, Equations (5) to (7) always decrease the loss value whenever the current variables do not have any infinite entries and the gradient is non-zero. Using notations $\bar{V}_i^{(k)} = \left[S_{i,1}^{(k)} V_1^{(k)T} \cdots S_{i,b}^{(k)} V_b^{(k)T}\right]^T$ and $\bar{U}_j^{(k)} = \left[(U_1^{(k+1)} S_{1,j}^{(k)})^T \cdots (U_b^{(k+1)} S_{b,j}^{(k)})^T\right]^T$ to indicate the concatenation of the right and left factors scaled by the diagonal components, the loss is monotonically non-increasing as in the following theorem.

**Theorem 1.** *Let $A_{i,j} \in \mathbb{R}^{p \times p}$ be a target block and $U_i^{(k)}, V_j^{(k)} \in \mathbb{R}^{p \times r}$, and $s_{i,j}^{(k)} \in \mathbb{R}^r$ be factors of a block in the BLAST matrix to be optimized. With the step sizes $0 < \eta_{U_i^{(k)}} \leq 1/\sigma_1\left(\bar{V}_i^{(k)T} \bar{V}_i^{(k)}\right)$, $0 < \eta_{V_j^{(k)}} \leq 1/\sigma_1\left(\bar{U}_j^{(k)T} \bar{U}_j^{(k)}\right)$, $0 < \eta_{s_{i,j}^{(k)}} \leq 1/\sigma_1\left((U_i^{(k+1)T} U_i^{(k+1)}) \odot (V_j^{(k+1)T} V_j^{(k+1)})\right)$, the gradient descent updates in Equations (5) to (7) monotonically non-increase the loss:*

$$\ell(U_*^{(k+1)}, V_*^{(k+1)}, s_{*,*}^{(k+1)}) \leq \ell(U_*^{(k)}, V_*^{(k)}, s_{*,*}^{(k)}).$$

The proof of Theorem 1 is in Section B, which is an application of the descent lemma from classical optimization theory.

**Blast Factorization via Preconditioned Gradient Descent**    Recall that in order to estimate the BLAST factors given a pre-trained weight $A$, we need to choose a rank $r$. Since we do not know the rank of $A$ a priori, we may have to overestimate the rank. However, overestimating the rank may slow down the convergence rate for solving Equation (4). To illustrate this, we performed an empirical analysis of the convergence behavior on a synthetically generated target low-rank matrix $A$, whose dimension is $256 \times 256$ with a true rank of $r^* = 8$. For the analysis, we computed the factors of a BLAST matrix with $b = 16$ for various values of $r$. We used linearly decaying step sizes $\eta^{(k)} = \eta_{U_i^{(k)}} = \eta_{V_j^{(k)}} = \eta_{s_{i,j}^{(k)}}$. When $r = r^* = 8$ (the ranks of the left and right factors $U_*, V_*$ match the actual rank of $A$), gradient descent finds a low-rank solution with minimal error within 30 iterations, as shown by the blue curve in Figure 3-left. However, in the case of $r = 32 > r^*$ where the BLAST factors are overparameterized, we observed slower convergence and a substantial residual

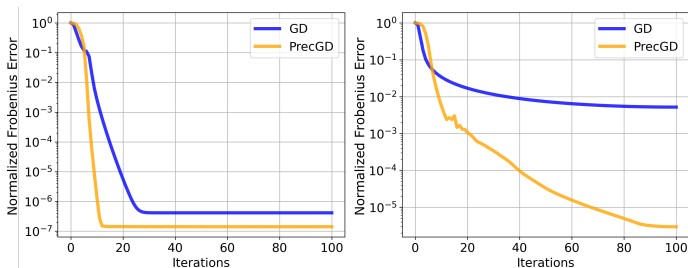

Figure 3: Convergence of the BLAST factorization with and without the preconditioning steps on noiseless low-rank matrix factorization with rank $r^\star$. Left: The BLAST parameter $r = r^\star$, Right: $r > r^\star$. When $r > r^\star$, the convergence rate of GD without the preconditioning is slowed down, while GD with the preconditioning (PrecGD) can recover the ground truth with small error.

---

**Algorithm 2** Preconditioned BLAST Factorization (see Equations (8) and (9))

---

**Require:** $\boldsymbol{A}$, $\{\eta^{(k)}\}_{k=0}^{K-1}, \epsilon, b, K, \delta > 0$

1: $\boldsymbol{U}_i^{(0)}, \boldsymbol{V}_j^{(0)} \sim \mathcal{N}(\boldsymbol{0}, \epsilon^2 \boldsymbol{I})$, $\boldsymbol{s}_{i,j}^{(0)} \sim \text{Unif}(0,1)$, $\qquad\qquad\qquad\qquad\qquad \forall i, j = 1, \ldots, b$

2: **for** $k = 0, 1, \ldots, K-1$ **do**

3: $\qquad \boldsymbol{U}_i^{(k+1)} \leftarrow \boldsymbol{U}_i^{(k)} - \eta^{(k)} \cdot \left( \boldsymbol{U}_i^{(k)} \bar{\boldsymbol{V}}_i^{(k)T} - \boldsymbol{A}_{i,*} \right) \bar{\boldsymbol{V}}_i^{(k)} \boldsymbol{P}_{\boldsymbol{U}_i}^{(k)}$, $\qquad\qquad\qquad \forall i = 1, \ldots, b$

4: $\qquad \boldsymbol{V}_j^{(k+1)} \leftarrow \boldsymbol{V}_j^{(k)} - \eta^{(k)} \cdot \left( \bar{\boldsymbol{U}}_j^{(k)} \boldsymbol{V}_j^{(k)T} - \boldsymbol{A}_{*,j} \right)^T \bar{\boldsymbol{U}}_j^{(k)} \boldsymbol{P}_{\boldsymbol{V}_j}^{(k)}$, $\qquad\qquad\qquad \forall j = 1, \ldots, b$

5: $\qquad \boldsymbol{s}_{i,j}^{(k+1)} \leftarrow \boldsymbol{s}_{i,j}^{(k)} - \eta^{(k)} \cdot \boldsymbol{P}_{\boldsymbol{s}_{i,j}}^{(k)} \left( \boldsymbol{W}_{i,j}^{(k)} \boldsymbol{s}_{i,j} - \text{diag} \left( \boldsymbol{U}_i^{(k+1)T} \boldsymbol{A}_{i,j} \boldsymbol{V}_j^{(k+1)} \right) \right), \forall i, j = 1, \ldots, b$

6: **end for**

7: **return** $\boldsymbol{U}_*^{(K)}, \boldsymbol{V}_*^{(K)}, \boldsymbol{s}_{*,*}^{(K)}$

---

error after 100 steps as shown by the blue curve in Figure 3-right. This behavior is consistent with previous observations of slow convergence in ill-conditioned matrix factorization problems [23, 24].

The convergence rate of solving the overparameterized low-rank factorization by gradient descent can be improved via inexpensive *preconditioners* [23, 24] which effectively decrease the condition number at each iteration. Inspired by the preconditioned gradient descent for low-rank factorization, we generalize the idea to solve our problem by multiplying preconditioning matrices to the gradients in Equations (5) to (7). We summarize the preconditioned gradient descent method for the BLAST factorization in Algorithm 2, where the following preconditioning matrices are used:

$$\boldsymbol{P}_{\boldsymbol{U}_i}^{(k)} = \left( \bar{\boldsymbol{V}}_i^{(k)T} \bar{\boldsymbol{V}}_i^{(k)} + \delta \boldsymbol{I} \right)^{-1}, \boldsymbol{P}_{\boldsymbol{V}_j}^{(k)} = \left( \bar{\boldsymbol{U}}_j^{(k)T} \bar{\boldsymbol{U}}_j^{(k)} + \delta \boldsymbol{I} \right)^{-1}, \tag{8}$$

$$\boldsymbol{P}_{\boldsymbol{s}_{i,j}}^{(k)} = \left( \boldsymbol{W}_{i,j}^{(k)} + \delta \boldsymbol{I} \right)^{-1}, \boldsymbol{W}_{i,j}^{(k)} = \left( \boldsymbol{U}_i^{(k+1)T} \boldsymbol{U}_i^{(k+1)} \right) \odot \left( \boldsymbol{V}_j^{(k+1)T} \boldsymbol{V}_j^{(k+1)} \right). \tag{9}$$

$\delta$ is proportional to the square root of the error in Equation (4). The derivations are presented in Appendix A.2. Figure 3 shows that preconditioning improves the convergence of the overparameterized BLAST factorization. The preconditioned gradient descent (yellow curve) finds the points with low error after 100 steps, whereas the gradient descent without preconditioning fails to achieve a small error. More empirical studies on the BLAST factorization with or without preconditioning can be found in Appendix D.1. We summarize the compression procedure in Algorithm 2. The computational complexity of this compression algorithm is $O(nr^2 + r^3)$ where the cubic dependency on $r$ is from the matrix inversion steps. We emphasize that these matrix inversion steps are not substantial computational bottlenecks because $r$ is smaller than $n$ as in prior work [23, 24].

After BLAST compression outlined in Algorithm 2, the BLAST factors can be used directly for inference to save both storage and computational costs. However, we observe that, instead of using the estimated factors directly, using them as initial points and refining the estimates by re-training the model with BLAST factors can further improve the performance of the compressed model. We refer to this process as "re-training" after BLAST compression.

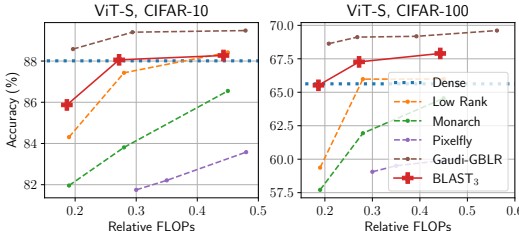

Figure 4: CIFAR-10/100 accuracy of ViT-S trained from scratch with different structured matrices.

| Model | Accuracy (%) | Relative FLOPs (%) |
|---|---|---|
| Dense ViT-Base | 78.7 | 100 |
| Low-Rank | 78.9 | 33.5 |
| Monarch [14] | 78.9 | 33.5 |
| Gaudi-GBLR [12] | 78.5 | 32.8 |
| $BLAST_3$ | **79.3** | **27.8** |

Table 1: ImageNet validation accuracy and relative FLOPs of ViT-Base trained from scratch models with different structured weight matrices. The image and the patch sizes are $224 \times 224$ and $16 \times 16$, respectively. $BLAST_3$ indicates the BLAST matrix with $3 \times 3$ number of blocks.

## 4 Experimental Results

We evaluate the BLAST matrix under two settings: (i) training from scratch with random initialization in the BLAST format, and (ii) re-training after compressing the dense weights to BLAST matrices via Algorithm 2. We compare the performance of BLAST with both non-adaptive and adaptive structured matrices. Among the non-adaptive approaches, we include low-rank (LR) matrices, Monarch for block low-rank (BLR) [14], and Pixelfly [17] or Block-Diagonal for block sparse matrices. For the adaptive and learnable structured matrix category, we evaluate Gaudi-GBLR [12]. We report the number of floating point operations (FLOPs) by counting the number of multiplications. The BLAST matrix with $b \times b$ number of blocks is denoted by $BLAST_b$. We used the same hyperparameter $r$ for every target weight matrix by setting it to meet the computational budget of the DNN. All experimental details can be found in Appendix C.

### 4.1 Training from Scratch

**Image Classification** We train the reduced-size Vision Transformers (ViT) [19] with $BLAST_3$ (BLAST with $b = 3$) weight matrices on CIFAR-10, 100 [28], and ImageNet-1k[29] for 310 epochs from *random initialization*, and compare with other structured matrices. In the CIFAR-10 and CIFAR-100 benchmarks, BLAST outperforms several non-adaptive baselines, such as Low-Rank, Pixelfly, and Monarch with higher accuracy at the same FLOPs complexity (Figure 4). Gaudi-GBLR presents the most favorable accuracy-to-FLOPs tradeoff due to its capability of learning the adaptive resource/budget allocation for each weight matrix, which is a feature that our BLAST setting lacks in this particular evaluation (as we force it to use the same $r$ for all matrices).

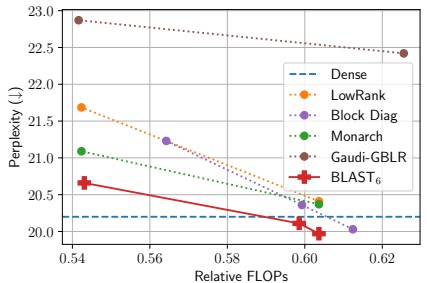

Figure 5: Pre-training result: WikiText 103 test perplexity-FLOPs trade-off curves from GPT-2 with different types of weight matrices.

However, in the context of ImageNet-1k in Table 1, weight matrices trained using BLAST with $b = 3$ attain the highest levels of accuracy with the least FLOPs. This superior performance of BLAST (despite the common $r$ for all matrices) over Gaudi-GBLR can be attributed to its simpler training process with fewer hyperparameters. In contrast, the more complex training requirements of Gaudi-GBLR, which involve smoothness annealing and proximal gradient descent, may lead to suboptimal results for a large model such as ViT-Base in Table 1.

**Language Model Evaluation** We validate the training performance of BLAST weights on language models. We replace the weights of GPT-2 [20] with random $BLAST_6$ matrices and trained the network from scratch on the WikiText 103 [30] dataset for 100 epochs. In Figure 5, we compare the test set perplexity of BLAST with the perplexity from low-rank, block-diagonal, Monarch, and Gaudi-GBLR matrices. Similar to the ImageNet training, we found that BLAST achieves the best perplexity-FLOPs trade-off. Compared to Gaudi-GBLR, BLAST obtains a significant perplexity gain. We attribute this

improvement to the simple training process of BLAST which requires less hyperparameter tuning than that of Gaudi-GBLR.

## 4.2 Compression and Re-training

In this section, we discuss the performance of BLAST weights when pre-trained dense weights are available. We first compress the dense weights using Algorithm 2 and re-train the model on the training data with the cross-entropy loss.

**ViT on ImageNet Classification** We compress the weights of the vision transformer (ViT) trained on ImageNet training set by $BLAST_3$ and $BLAST_{12}$ using Algorithm 2 and re-train the models for 35 epochs. The accuracy-FLOPs trade-off curve of each model is presented in Figure 6. Both BLAST compressed & re-trained models outperform other baselines, even though BLAST models did not use the adaptive budget allocation, unlike Gaudi-GBLR. It is observed that the accuracy of the BLAST models slightly increases from $b = 3$ to $b = 12$.

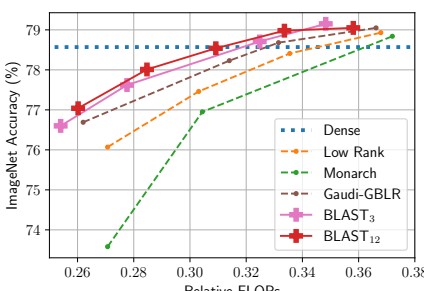

Figure 6: Compression and re-training result: ImageNet accuracy-FLOPs trade-off curves from ViT-B with different types of weight matrices.

**Diffusion Models** We compress the weights of a Diffusion Transformer (DiT) [13] pre-trained on ImageNet using BLAST matrices and compare its performance to SVD-based low-rank approximation. For both techniques, we match the compression ratio such that both decrease the total number of model parameters by 50%, and re-train each model for 10 epochs on the ImageNet training set. We evaluate the models by generating a total of 50,000 images using the original, low-rank, and BLAST compressed models, and compute the FID [31], sFID [32], and IS [33] metrics with respect to the ImageNet validation set. The objective is to observe if the compressed model can generate images as realistic as the original uncompressed model.

| CR | Method | FID($\downarrow$) | sFID($\downarrow$) | IS($\uparrow$) |
|---|---|---|---|---|
| 0% | Original | 9.62 | 6.85 | 121.50 |
| 50% | Low-Rank | 48.07 | 11.44 | 26.09 |
| | $BLAST_9$ | 10.45 | 6.72 | 111.05 |

Table 2: Performance comparison for compressing the weight matrices of a diffusion model followed by re-training. FID and IS scores were computed with respect to a validation dataset. CR stands for Compression Ratio.

In Table 2, we show quantitatively that the model compressed via BLAST significantly outperforms the model compressed via SVD. The low-rank compressed model often generates unrealistic images, leading to poor metrics such as the inception score. Figure 1 also contrasts how the BLAST matrices contribute to maintaining high perceptual quality as well as a close instance-wise resemblance with the uncompressed model outputs. Due to space limitations, we defer additional qualitative results and experimental setup to Appendix D.2.

**Large Language Models (LLMs)** We compress the weights of Llama-7B [1] with BLAST matrices using Algorithm 2 by 20% and 50%, and re-train the models for 400 steps on a subset of SlimPajama [34] dataset using 0.49B tokens. The number of blocks $b$ in the BLAST matrices is fixed at 16, and we use $r = 1024$ for the attention modules and $r = 1488$ for the MLP modules to achieve a 50% compression ratio. We test the WikiText-2 perplexity and the zero-shot task classification accuracy on common sense reasoning datasets including PIQA[35], HellaSwag[36], WinoGrande[37], BoolQ[38], OpenBookQA[39], ARC-easy and challenge [40]. We report the performance of Low-Rank, Monarch, and Block-Diagonal weight matrices after compression at the same rate and re-training. In Table 3, the first row presents the performance of the original Llama-7B model. On 50% compression ratio in the last five rows, the Monarch and Block-Diagonal matrices fail to recover the acceptable performance. Compared to Low-Rank weights, BLAST weights achieve the lowest performance degradation in WikiText-2 perplexity and zero-shot classification accuracy. The accuracy of each common sense reasoning benchmark and extended results can be found in Appendix D.3.

We provide an analysis to quantify the performance impact of compression and re-training. We first quantify the weight compression performance at 20% compression ratio in Table 3. Although the compression ratio is moderate, Low-Rank and Monarch compression without re-training suffer from

| CR | Method | # Params | Re-trained? | WikiText-2 Perplexity (↓) | Avg. 0-Shot Accuracy (%) (↑) |
|---|---|---|---|---|---|
| 0% | Original Llama-7B | 6.74B | N/A | 9.37 | 66.07 |
| 20% | Low-Rank | 5.41B | No | 23.67 (-14.30) | 59.57 (-6.50) |
| | Monarch [14] (BLR) | 5.41B | No | 47.18 (-37.81) | 48.91(-17.17) |
| | BLAST$_{16}$ | 5.41B | No | 12.13 (-2.76) | 62.94 (-3.14) |
| 50% | Low-Rank | 3.51B | Yes | 26.33 (-16.96) | 48.40 (-17.67) |
| | Monarch [14] (BLR) | 3.50B | Yes | 7.53e5 (-7.53e5) | 35.03 (-31.04) |
| | Block-Diagonal | 3.50B | Yes | 5.21e6 (-5.21e6) | 34.86 (-31.21) |
| | BLAST$_{16}$ | 3.56B | Yes | 14.21 (-4.84) | **56.22 (-9.84)** |

Table 3: Zero-shot performance of LLaMA-7B after compression and retraining. Avg. 0-Shot Accuracy stands for the average accuracy of the zero-shot classification task. CR denotes compression ratio. **Bold** indicates the best performance under the same compression ratio. BLAST$_b$ indicates the BLAST matrix with $b \times b$ number of blocks.

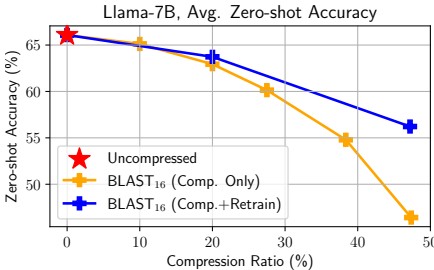

Figure 7: Average zero-shot accuracy vs. compression ratio curves of Llama-7B by BLAST$_{16}$ before and after re-training.

| CR | $b$ | $L = 10$ | $L = 100$ | $L = 1000$ |
|---|---|---|---|---|
| 0% | N/A | 0.41 ±8e-5 | 3.82 ±9e-4 | 41.23 ±6e-3 |
| 20% | 2 | 0.35 ±9e-5 | 3.30 ±2e-3 | 35.99 ±4e-3 |
| 20% | 16 | 0.36 ±8e-5 | 3.36 ±2e-3 | 36.48 ±7e-3 |
| 50% | 16 | 0.31 ±4e-4 | 2.86 ±1e-2 | 30.35 ±2e-2 |

Table 4: Average runtime (in second) of Llama-7B with BLAST$_b$ weights from 10 runs of text generation. ±: standard deviation, $L$: the length of the generated sequence, CR: Compression Ratio. All models evaluated on a single A100 GPU (40GB) using PyTorch [21] after `torch.compile()`.

significant performance loss, at most 4x higher perplexity and 25% accuracy degradation. On the other hand, BLAST$_{16}$ compression without re-training maintains reasonable accuracy and perplexity. This shows that the flexible and adaptive structure of BLAST matrices captures more information than other types of structured matrices. Yet, BLAST compression also exhibits noticeable performance degradation on a more intensive compression ratio (see the yellow curve in Figure 7). We provide more compression-only results on Diffusion Models and LLMs in Appendix D.

We find that the re-training stage is crucial for converting a pre-trained model into an efficient version without losing significant accuracy when the compression ratio is high. In Figure 7, we show the average zero-shot classification accuracy of the compressed models with 50% compression. The models with BLAST weights before re-training (yellow curve) exhibit substantial accuracy degradation at higher compression ratios. However, re-training (blue curve) effectively recovers performance using only 0.49B tokens and 400 steps.

**LLM Runtime Analysis** We evaluate the runtime of the Llama-7B compressed by the BLAST matrices on the text generation task. For evaluation, we let the model generate the sequences of length $L =$10, 100, and 1000 ten times and report the average and the standard deviation of model runtime in Table 4. The instruction we use to generate the desired sequence length is "`Increasing sequence: one,`" and the model generates the text sequence such as "`two, three, `" and so on, with a batch size of 1. All runtime evaluation tests were conducted on a single 40GB NVIDIA A100 GPU after compiling the script using `torch.compile()`. The 20% compressed model shows a 12%~15% runtime reduction without any library function customization for BLAST matrix multiplication. The speedup when $b = 2$ is higher than when $b = 16$ because a larger number of blocks increases the computation overhead to perform Equation (3). Notably, the 50% compression ratio provides 32%~35% runtime reduction when $b = 16$. We note that the test is highly memory-bandwidth-bounded. Thus the speedup reported in Table 4 can be mostly attributed to the reduction of parameters (i.e., memory accesses) rather than the reduction in FLOPs due to BLAST compression.

## 5 Related Works

**Structured Matrix with Shared Bases** Sharing the bases of block-structured matrices has recently drawn interest due to its considerable memory savings. BLAST matrices exemplify this approach.

Ashcraft et al. [41] extend the BLR [15] format to BLR$^2$, incorporating shared bases and block-wise low-rank coupling factors determined through LQ or QR factorization. Similarly, Yen et al. [42] apply the concept of shared bases in the Block-Diagonal preconditioner for DNN weights. While BLAST also shares the bases of blocks, it is distinct in having a diagonal coupling matrix, as shown in Equation (2). The design of BLAST matrices aims to enhance efficiency and learn a variety of structures, from low-rank to high-rank block matrices. More importantly, identifying the diagonal coupling factors in BLAST matrices does not necessitate QR decomposition. Instead, they can be updated via gradient descent, making this approach well-suited for modeling the weight matrices in deep learning models.

**DNN Weight Pruning and Decomposition** Earlier work on DNN pruning [43–45] identifies less important parameters from the Hessian or magnitude to sparsify the weight matrix. Unlike general sparse matrices, a group sparse matrix skips computation in a group-wise manner by pruning channels [46, 47]. Sparse GPT [11] successfully prunes large language models with 50% sparsity without significantly degrading the performance of the original model. The model can achieve actual speedup utilizing 2:4 sparsity [48] on specific GPUs. However, 2:4 sparsity requires accelerators with specific architectures (e.g., NVIDIA A100 GPUs) and supports only the 50% compression ratio. On the other hand, BLAST is device-agnostic since it can be implemented with basic matrix arithmetic operations and offers diverse compression ratios.

Low-rank matrices have been widely adopted for CNN compression [49, 50] and Transformer compression [51, 52]. Additionally, Butterfly [53] and Monarch [14] factorization methods model high-rank but low-dimensional structures of the weights. Specifically, a Monarch matrix is a generalized version of a Butterfly matrix, yielding a wider spectrum of structured matrices. The number of blocks plays a key role in determining the rank of the Monarch matrix as a whole and does not generalize to another Monarch matrix with fewer blocks. Unlike Monarch, BLAST with $b \times b$ blocks can express Monarch matrices with the same or fewer number of blocks, including the global low-rank matrix, i.e., $b = 1$.

**Learning Low-dimensional Structures of DNN Weights** Similar to the BLAST matrix, a Gaudi-GBLR matrix [12] enables learning low-dimensional structures by gradient descent in the generalized structured matrix space. Gaudi-GBLR overlaps a variable number of zero-padded rank-1 blocks to model high-rank submatrices. Although Gaudi-GBLR can express a wider spectrum of matrices than BLAST, the matrix-vector multiplication for Gaudi-GBLR is less efficient because GPUs and typical neural processors cannot handle zero-padded vectors efficiently. In contrast, the BLAST matrix-vector operation does not involve zero padding, allowing for more efficient execution in hardware for the same FLOPs (as shown in Figure 4, Figure 6, and Table 1).

# 6 Conclusion and Future Work

In this work, we introduced the BLAST matrix designed to improve the inference efficiency of large DNNs. The BLAST matrix represents various low-dimensional structures of the weight matrices with fewer parameters, while enabling efficient matrix-vector products. The BLAST factors are either learnable from data or estimated from existing weights using our preconditioned factorization algorithm. Our results on both language and vision tasks highlight the effectiveness of BLAST.

**Limitations and Future Work** The BLAST matrix-vector product consists of three steps, as detailed in Equation (3), which may degrade hardware-execution parallelism. In our evaluation, we used the same computational budget $r$ for all matrices. Learning an adaptive budget per layer or matrix (e.g., via overparameterization [54, 7]) could further improve BLAST performance, which is left for future work. The proposed method has not been evaluated on tiny (<100M parameters) or extremely large (>10B parameters) DNNs. Additionally, optimizing runtime and power consumption via BLAST matrices with customized library functions and/or hardware accelerators also remains as future work. Furthermore, a deeper theoretical investigation into the behaviors of BLAST matrices would provide a more comprehensive understanding of their capabilities and limitations. Applying advanced re-training techniques, such as knowledge distillation [55] or iterative compression and distillation [56], to the BLAST compression pipeline is also left for future work. Finally, beyond the weight structures, we expect BLAST can also help understand and exploit low-dimensional *data* manifolds [57–59] in future work.

## Acknowledgments and Disclosure of Funding

The arXiv version of the paper can be found at https://arxiv.org/abs/2410.21262. This work was supported in part by COGNISENSE, one of seven centers in JUMP 2.0, a Semiconductor Research Corporation (SRC) program sponsored by DARPA. SMK and QQ acknowledge funding support from NSF CAREER CCF-2143904, and NSF CCF-2212066. We thank Salar Fattahi, Reetuparna Das, Mingyu Yang, Sara Shoouri, Shrikant Arvavasu, Jayeon Yi, Andrew Larson, Pierre Abillama, Alireza Khadem, Yufeng Gu, and Can Yaras for helpful discussion and feedback.

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

# A Details on BLAST Matrix and Factorization

**Matrix Multiplication Code Implementation** Following the conventional setting in Transformers [16], the input tensor is assumed to have batch, sequence, and channel dimensions. The left and right factors are multiplied using the batched matrix multiplication routine, whereas the diagonal factors are multiplied via broadcasting and summation.

```
def blast_matmul(
    X,   # shape=(B, n, q*b), B=batch_size, n=num_seq
    U,   # shape=(b, p, r), b=num_blocks, r=BLAST rank
    S,   # shape=(b, b, r)
    Vt,  # shape=(b, r, q)
):
    X = rearrange(X, "B n (q b) -> b (B n) q")
    Y = bmm(X, Vt.T)  # multiply right factor
    Z = Y.unsqueeze(0) * S.unsqueeze(2)   # multiply diagonal factor
    Z = Z.sum(1)  # aggregate, shape=(b, B*n, r)
    Out = bmm(Z, U.T)  # multiply left factor
    Out = rearrange(Out, "b (B n) p -> B n (b p)")
    return Out
```

Figure 8: Pseudocode of BLAST Matrix Multiplication. The function `bmm` stands for the batched matrix multiplication routine (e.g., `torch.bmm` [21]).

## A.1 More Special Cases of BLAST Matrix

**Block diagonal matrix** A block-diagonal matrix is a BLAST matrix when $r = p$ and $s_{i,j} = \begin{cases} \mathbf{1}_r & \text{if } i = j \\ \mathbf{0}_r & \text{otherwise} \end{cases}$ since

$$\begin{bmatrix} \boldsymbol{A}_{1,1} & & & \\ & \boldsymbol{A}_{2,2} & & \\ & & \ddots & \\ & & & \boldsymbol{A}_{b,b} \end{bmatrix} = \begin{bmatrix} \boldsymbol{U}_1\mathrm{diag}(\boldsymbol{s}_{1,1})\boldsymbol{V}_1^T & & & \\ & \boldsymbol{U}_2\mathrm{diag}(\boldsymbol{s}_{2,2})\boldsymbol{V}_2^T & & \\ & & \ddots & \\ & & & \boldsymbol{U}_b\mathrm{diag}(\boldsymbol{s}_{b,b})\boldsymbol{V}_b^T \end{bmatrix}.$$

When $r < p$, BLAST matrices model block-diagonal matrices which have low-rank diagonal blocks.

**Block low-rank (BLR) matrix** For ease of understanding, let us consider a BLR matrix of 9 $\frac{n}{3} \times \frac{n}{3}$ rank-1 blocks. Each block is composed of the unique bases $\boldsymbol{A}_{i,j} = \boldsymbol{u}_{i,j}\boldsymbol{v}_{i,j}^T$. Now consider $\boldsymbol{U}_i = [\boldsymbol{u}_{i,1}, \boldsymbol{u}_{i,2}, \boldsymbol{u}_{i,3}]$ and $\boldsymbol{V}_j = [\boldsymbol{v}_{1,j}, \boldsymbol{v}_{2,j}, \boldsymbol{v}_{3,j}]$. Then, the BLR matrix is a BLAST matrix with $r = b = 3$:

$$\begin{bmatrix} \boldsymbol{u}_{1,1}\boldsymbol{v}_{1,1}^T & \boldsymbol{u}_{1,2}\boldsymbol{v}_{1,2}^T & \boldsymbol{u}_{1,3}\boldsymbol{v}_{1,3}^T \\ \boldsymbol{u}_{2,1}\boldsymbol{v}_{2,1}^T & \boldsymbol{u}_{2,2}\boldsymbol{v}_{2,2}^T & \boldsymbol{u}_{2,3}\boldsymbol{v}_{2,3}^T \\ \boldsymbol{u}_{3,1}\boldsymbol{v}_{3,1}^T & \boldsymbol{u}_{3,2}\boldsymbol{v}_{3,2}^T & \boldsymbol{u}_{3,3}\boldsymbol{v}_{3,3}^T \end{bmatrix} = \begin{bmatrix} \boldsymbol{U}_1\boldsymbol{S}_1\boldsymbol{V}_1^T & \boldsymbol{U}_1\boldsymbol{S}_2\boldsymbol{V}_2^T & \boldsymbol{U}_1\boldsymbol{S}_3\boldsymbol{V}_b^T \\ \boldsymbol{U}_2\boldsymbol{S}_1\boldsymbol{V}_1^T & \boldsymbol{U}_2\boldsymbol{S}_2\boldsymbol{V}_2^T & \boldsymbol{U}_2\boldsymbol{S}_3\boldsymbol{V}_3^T \\ \boldsymbol{U}_3\boldsymbol{S}_1\boldsymbol{V}_1^T & \boldsymbol{U}_3\boldsymbol{S}_2\boldsymbol{V}_2^T & \boldsymbol{U}_3\boldsymbol{S}_3\boldsymbol{V}_3^T \end{bmatrix}$$

where $\boldsymbol{S}_1 = \mathrm{diag}([1, 0, 0])$, $\boldsymbol{S}_2 = \mathrm{diag}([0, 1, 0])$, and $\boldsymbol{S}_3 = \mathrm{diag}([0, 0, 1])$.

To model a $n \times n$ general $b \times b$ partitioned BLR matrix where the rank of each block is $t$, let us use the BLAST matrix with $b \times b$ blocks and $r = bt$. The factors have the following shapes:

$$\boldsymbol{U}_i, \boldsymbol{V}_j \in \mathbb{R}^{p \times (bt)}, \quad \boldsymbol{s}_{i,j} \in \mathbb{R}^{bt}.$$

By letting $s_{i,j,k} = \begin{cases} 1 & \text{if } t(j-1) + 1 \le k < tj + 1 \\ 0 & \text{otherwise} \end{cases}$, the BLAST matrix can model the BLR matrix.

Note that the number of parameters of the BLAST matrix is $2nr + rb^2$, whereas that of the BLR matrix in this case is $b^2 \cdot (p + p)t = 2(pb)(bt) = 2nr$. In other words, the BLAST matrix models various matrices with the cost of $rb^2$.

## A.2 Derivation of Preconditioning Matrices

We rewrite the loss function in Equation (4) below:

$$\ell(\boldsymbol{U}_*, \boldsymbol{V}_*, \boldsymbol{s}_{*,*}) = \sum_{i=1}^{b} \sum_{j=1}^{b} \frac{1}{2} \left\| \boldsymbol{A}_{i,j} - \boldsymbol{U}_i \text{diag}(\boldsymbol{s}_{i,j}) \boldsymbol{V}_j^T \right\|_F^2. \tag{4}$$

We first derive the gradients of $\boldsymbol{U}_i$, $\boldsymbol{V}_j$, and $\boldsymbol{s}_{i,j}$, then discuss the preconditioning matrix for each factor.

### A.2.1 Gradients

Here we derive the gradients of Equation (4) with respect to the BLAST factors. We begin with introducing the short-handed notation for the concatenated factors:

$$\bar{\boldsymbol{V}}_i^T = [\boldsymbol{S}_{i,1}\boldsymbol{V}_1^T \cdots \boldsymbol{S}_{i,b}\boldsymbol{V}_b^T],$$
$$\bar{\boldsymbol{U}}_j = [(\boldsymbol{U}_1\boldsymbol{S}_{1,j})^T \cdots (\boldsymbol{U}_b\boldsymbol{S}_{b,j})^T]^T.$$

That is, the matrix $\bar{\boldsymbol{V}}_i$ is composed by concatenating $\boldsymbol{V}_j^T$s horizontally along $j = 1, 2, \ldots, b$ after scaling them with $\boldsymbol{S}_{i,j}$. $\bar{\boldsymbol{U}}_j$ is defined similarly by concatenating the scaled $\boldsymbol{U}_i$s vertically.

Now we derive the gradients below.

**Gradient of $\boldsymbol{U}_i$** We only have to consider the loss term related to $\boldsymbol{U}_i$. Therefore, we have the following gradient expression:

$$
\begin{aligned}
\nabla_{\boldsymbol{U}_i} \ell(\boldsymbol{U}_*, \boldsymbol{V}_*, \boldsymbol{s}_{*,*}) &= \nabla_{\boldsymbol{U}_i} \sum_{j=1}^{b} \frac{1}{2} \left\| \boldsymbol{A}_{i,j} - \boldsymbol{U}_i \text{diag}(\boldsymbol{s}_{i,j}) \boldsymbol{V}_j^T \right\|_F^2 \\
&= \nabla_{\boldsymbol{U}_i} \frac{1}{2} \left\| \boldsymbol{A}_{i,*} - \boldsymbol{U}_i \bar{\boldsymbol{V}}_i^T \right\|_F^2 \\
&= (\boldsymbol{U}_i \bar{\boldsymbol{V}}_i^T - \boldsymbol{A}_{i,*}) \bar{\boldsymbol{V}}_i,
\end{aligned}
\tag{10}
$$

where for the second equality we used the concatenated version of the first line.

**Gradient of $\boldsymbol{V}_j$** follows the similar derivation as Equation (10):

$$
\begin{aligned}
\nabla_{\boldsymbol{V}_j} \ell(\boldsymbol{U}_*, \boldsymbol{V}_*, \boldsymbol{s}_{*,*}) &= \nabla_{\boldsymbol{V}_j} \sum_{i=1}^{b} \frac{1}{2} \left\| \boldsymbol{A}_{i,j} - \boldsymbol{U}_i \text{diag}(\boldsymbol{s}_{i,j}) \boldsymbol{V}_j^T \right\|_F^2 \\
&= \nabla_{\boldsymbol{V}_j} \frac{1}{2} \left\| \boldsymbol{A}_{*,j} - \bar{\boldsymbol{U}}_j \boldsymbol{V}_j^T \right\|_F^2 \\
&= (\bar{\boldsymbol{U}}_j \boldsymbol{V}_j^T - \boldsymbol{A}_{*,j})^T \bar{\boldsymbol{U}}_j.
\end{aligned}
\tag{11}
$$

**Gradient of $\boldsymbol{s}_{i,j}$** We consider the block-wise loss for the gradient:

$$\nabla_{\boldsymbol{s}_{i,j}} \ell(\boldsymbol{U}_*, \boldsymbol{V}_*, \boldsymbol{s}_{*,*}) = \nabla_{\boldsymbol{s}_{i,j}} \frac{1}{2} \left\| \boldsymbol{A}_{i,j} - \boldsymbol{U}_i \text{diag}(\boldsymbol{s}_{i,j}) \boldsymbol{V}_j^T \right\|_F^2. \tag{12}$$

Since the Frobenius norm can be expressed by a matrix trace, the loss is written as follows:

$$
\begin{aligned}
\left\| \boldsymbol{A}_{i,j} - \boldsymbol{U}_i \text{diag}(\boldsymbol{s}_{i,j}) \boldsymbol{V}_j^T \right\|_F^2 &= \text{Tr}\left( \left( \boldsymbol{A}_{i,j} - \boldsymbol{U}_i \boldsymbol{S}_{i,j} \boldsymbol{V}_j^T \right)^T \left( \boldsymbol{A}_{i,j} - \boldsymbol{U}_i \boldsymbol{S}_{i,j} \boldsymbol{V}_j^T \right) \right) \\
&= \text{Tr}\left( \boldsymbol{V}_j \boldsymbol{S}_{i,j} \boldsymbol{U}_i^T \boldsymbol{U}_i \boldsymbol{S}_{i,j} \boldsymbol{V}_j^T - 2\boldsymbol{A}_{i,j}^T \boldsymbol{U}_i \boldsymbol{S}_{i,j} \boldsymbol{V}_j^T + \boldsymbol{A}_{i,j}^T \boldsymbol{A}_{i,j} \right) \\
&= \text{Tr}\left( \boldsymbol{S}_{i,j} \boldsymbol{V}_j^T \boldsymbol{V}_j \boldsymbol{S}_{i,j} \boldsymbol{U}_i^T \boldsymbol{U}_i - 2\boldsymbol{S}_{i,j} \boldsymbol{V}_j^T \boldsymbol{A}_{i,j}^T \boldsymbol{U}_i + \boldsymbol{A}_{i,j}^T \boldsymbol{A}_{i,j} \right),
\end{aligned}
$$

where $\text{Tr}(\boldsymbol{X})$ is the trace of $\boldsymbol{X}$. Note that the derivative of product in trace is given by $\nabla_{\boldsymbol{X}} \text{Tr}(\boldsymbol{XY}) = \boldsymbol{Y}^T$ for any two conformal matrices $\boldsymbol{X}$ and $\boldsymbol{Y}$. Therefore, we have

$$\nabla_{\boldsymbol{S}_{i,j}} \frac{1}{2} \left\| \boldsymbol{A}_{i,j} - \boldsymbol{U}_i \text{diag}(\boldsymbol{s}_{i,j}) \boldsymbol{V}_j^T \right\|_F^2 = \boldsymbol{U}_i^T \boldsymbol{U}_i \boldsymbol{S}_{i,j} \boldsymbol{V}_j^T \boldsymbol{V}_j - \boldsymbol{U}_i^T \boldsymbol{A}_{i,j} \boldsymbol{V}_j.$$

Now Equation (12) becomes as follows:

$$\nabla_{\boldsymbol{s}_{i,j}} \frac{1}{2} \left\| \boldsymbol{A}_{i,j} - \boldsymbol{U}_i \mathrm{diag}(\boldsymbol{s}_{i,j}) \boldsymbol{V}_j^T \right\|_F^2 = \mathrm{diag}\left( \boldsymbol{U}_i^T \boldsymbol{U}_i \boldsymbol{S}_{i,j} \boldsymbol{V}_j^T \boldsymbol{V}_j - \boldsymbol{U}_i^T \boldsymbol{A}_{i,j} \boldsymbol{V}_j \right). \qquad (13)$$

The first term on the right hand side is further arranged by using the fact that $\mathrm{diag}\left( \boldsymbol{X} \boldsymbol{Y}^T \right) = (\boldsymbol{X} \odot \boldsymbol{Y}) \mathbf{1}$ for any two matrices $\boldsymbol{X}, \boldsymbol{Y}$ of the same size.

$$\begin{aligned}
\mathrm{diag}\left( \left( \boldsymbol{U}_i^T \boldsymbol{U}_i \boldsymbol{S}_{i,j} \right) \left( \boldsymbol{V}_j^T \boldsymbol{V}_j \right) \right) &= \left[ \left( \boldsymbol{U}_i^T \boldsymbol{U}_i \mathrm{diag}(\boldsymbol{s}_{i,j}) \right) \odot \left( \boldsymbol{V}_j^T \boldsymbol{V}_j \right) \right] \mathbf{1}_r \\
&= \left( \left( \boldsymbol{U}_i^T \boldsymbol{U}_i \right) \odot \left( \boldsymbol{V}_j^T \boldsymbol{V}_j \right) \right) \boldsymbol{s}_{i,j}.
\end{aligned} \qquad (14)$$

Hence, the gradient of $\boldsymbol{s}_{i,j}$ is now expressed as follows:

$$\nabla_{\boldsymbol{s}_{i,j}} \ell(\boldsymbol{U}_*, \boldsymbol{V}_*, \boldsymbol{s}_{*,*}) = \left( \left( \boldsymbol{U}_i^T \boldsymbol{U}_i \right) \odot \left( \boldsymbol{V}_j^T \boldsymbol{V}_j \right) \right) \boldsymbol{s}_{i,j} - \mathrm{diag}\left( \boldsymbol{U}_i^T \boldsymbol{A}_{i,j} \boldsymbol{V}_j \right). \qquad (15)$$

### A.2.2 Preconditioning Matrices

Now let us derive the preconditioning matrices used in Algorithm 2.

**Preconditioning matrix $\boldsymbol{P}_{U_i}$ for $\boldsymbol{U}_i$** Let $\boldsymbol{V}_j$ and $\boldsymbol{s}_{i,j}$ are given. Let us consider the case when $\boldsymbol{U}_i$ is at the stationary point $\hat{\boldsymbol{U}}$ which satisfies

$$\begin{aligned}
\nabla_{\hat{\boldsymbol{U}}} \frac{1}{2} \| \boldsymbol{A}_{i,*} - \hat{\boldsymbol{U}} \bar{\boldsymbol{V}}_i^T \|_F^2 &= \hat{\boldsymbol{U}} \bar{\boldsymbol{V}}_i^T \bar{\boldsymbol{V}}_i - \boldsymbol{A}_{i,*} \bar{\boldsymbol{V}}_i \\
&= \boldsymbol{O},
\end{aligned}$$

where $\boldsymbol{O}$ is the zero matrix. This gives us the normal equation

$$\boldsymbol{A}_{i,*} \bar{\boldsymbol{V}}_i = \hat{\boldsymbol{U}} \bar{\boldsymbol{V}}_i^T \bar{\boldsymbol{V}}_i. \qquad (16)$$

Now consider a preconditioned gradient descent with $\boldsymbol{P}_{U_i} \in \mathbb{R}^{r \times r}$:

$$\begin{aligned}
\boldsymbol{U}_i' &= \boldsymbol{U}_i - \left( \boldsymbol{U}_i \bar{\boldsymbol{V}}_i^T \bar{\boldsymbol{V}}_i - \boldsymbol{A}_{i,*} \bar{\boldsymbol{V}}_i \right) \boldsymbol{P}_{U_i} \\
&= \boldsymbol{U}_i - \left( \boldsymbol{U}_i \bar{\boldsymbol{V}}_i^T \bar{\boldsymbol{V}}_i - \hat{\boldsymbol{U}} \bar{\boldsymbol{V}}_i^T \bar{\boldsymbol{V}}_i \right) \boldsymbol{P}_{U_i} \\
&= \boldsymbol{U}_i - \left( \boldsymbol{U}_i - \hat{\boldsymbol{U}} \right) \bar{\boldsymbol{V}}_i^T \bar{\boldsymbol{V}}_i \boldsymbol{P}_{U_i} \\
\implies \boldsymbol{U}_i' - \hat{\boldsymbol{U}}_i &= \left( \boldsymbol{U}_i - \hat{\boldsymbol{U}} \right) \left( \boldsymbol{I} - \bar{\boldsymbol{V}}_i^T \bar{\boldsymbol{V}}_i \boldsymbol{P}_{U_i} \right)
\end{aligned}$$

Suppose $\bar{\boldsymbol{V}}_i^T \bar{\boldsymbol{V}}_i$ is invertible. Then the ideal preconditioner is

$$\boldsymbol{P}_{U_i}^{\star} = (\bar{\boldsymbol{V}}_i^T \bar{\boldsymbol{V}}_i)^{-1} \qquad (17)$$

since it brings $\boldsymbol{U}_i'$ to the stationary point $\hat{\boldsymbol{U}}$. However, directly use the inverse of $\bar{\boldsymbol{V}}_i^T \bar{\boldsymbol{V}}$ might result in numerical instability or complete breakdown of the algorithm when the matrix is singular. By following [24], we use the regularized version

$$\boldsymbol{P}_{U_i} = \left( \bar{\boldsymbol{V}}_i^T \bar{\boldsymbol{V}}_i + \delta \boldsymbol{I} \right)^{-1} \qquad (18)$$

where $\delta$ is chosen by

$$\delta = \delta_0 \cdot \sqrt{\ell(\boldsymbol{U}_*, \boldsymbol{V}_*, \boldsymbol{s}_{*,*})}, \quad \delta_0 > 0. \qquad (19)$$

**Preconditioning matrix $\boldsymbol{P}_{V_j}$ for $\boldsymbol{V}_j$** The preconditioner for $\boldsymbol{V}_j$ can be derived by following the similar steps for $\boldsymbol{P}_{U_i}$. Here we present the result:

$$\boldsymbol{P}_{V_j} = \left( \bar{\boldsymbol{U}}_j^T \bar{\boldsymbol{U}}_j + \delta \boldsymbol{I} \right)^{-1} \qquad (20)$$

where $\delta$ is chosen by Equation (19).

**Preconditioning matrix $P_{s_{i,j}}$ for $s_{i,j}$**   We again consider the stationary point $\hat{s}$ or the diagonal version $\hat{S} = \text{diag}(\hat{s})$ which has a zero gradient:

$$\nabla_{\hat{s}} \frac{1}{2} \|A_{i,j} - U_i \hat{S} V_j^T\|_F^2 = \left((U_i^T U_i) \odot (V_j^T V_j)\right) \hat{s} - \text{diag}\left(U_i^T A_{i,j} V_j\right) = \mathbf{0}$$

$$\implies \left((U_i^T U_i) \odot (V_j^T V_j)\right) \hat{s} = \text{diag}\left(U_i^T A_{i,j} V_j\right).$$

Therefore, Equation (15) can be written as follows:

$$\nabla_{s_{i,j}} \ell(U_*, V_*, s_{*,*}) = \left((U_i^T U_i) \odot (V_j^T V_j)\right)(s_{i,j} - \hat{s}).$$

Now consider the preconditioned gradient descent:

$$s'_{i,j} = s_{i,j} - P_{s_{i,j}} \left((U_i^T U_i) \odot (V_j^T V_j)\right)(s_{i,j} - \hat{s})$$

$$\implies s'_{i,j} - \hat{s} = s_{i,j} - \hat{s} - P_{s_{i,j}} \left((U_i^T U_i) \odot (V_j^T V_j)\right)(s_{i,j} - \hat{s})$$

$$= \left(I - P_{s_{i,j}} \left((U_i^T U_i) \odot (V_j^T V_j)\right)\right)(s_{i,j} - \hat{s})$$

The Hadamard product of two positive definite matrices are still positive definite (see [60, Theorem 7.5.3], also known as Schur's Product Theorem). Hence, if both $U_i^T U_i$ and $V_j^T V_j$ are positive definite so that invertible, $(U_i^T U_i) \odot (V_j^T V_j)$ is also invertible. The ideal preconditioner when both matrices are invertible is therefore

$$P_{s_{i,j}}^\star = \left((U_i^T U_i) \odot (V_j^T V_j)\right)^{-1}. \tag{21}$$

Same as for $P_{U_i}$ and $P_{V_j}$, we also consider the regularized version:

$$P_{s_{i,j}} = \left((U_i^T U_i) \odot (V_j^T V_j) + \delta I\right)^{-1}. \tag{22}$$

Here, $\delta$ is chosen by Equation (19).

# B   Proof of Theorem 1

We first introduce the following properties:

**Lemma 2.** *[61, Lemma 3.4] Assume $f : \mathbb{R}^n \to \mathbb{R}$ is convex and continuously differentiable, and its gradient is $L$-Lipschitz continuous. Then for any $x, y \in \mathbb{R}^n$, one has*

$$f(y) - f(x) - \langle \nabla f(x), y - x \rangle \le \frac{L}{2} \|y - x\|_2^2.$$

*Proof.* See [61, Lemma 3.4]. $\qquad\qquad\qquad\qquad\qquad\qquad\qquad\qquad\qquad\qquad\qquad\qquad \square$

**Lemma 3.** *Assume $f : \mathbb{R}^n \to \mathbb{R}$ is convex and continuously differentiable, and its gradient is $L$-Lipschitz continuous. Consider a gradient descent update*

$$x^{(k+1)} = x^{(k)} - \eta \cdot \nabla f(x^{(k)}).$$

*Then, with the step size $0 < \eta \le \frac{1}{L}$, the following holds:*

$$f(x^{(k+1)}) \le f(x^{(k)}) - \frac{1}{2L} \|\nabla f(x^{(k)})\|_2^2.$$

*That is, the gradient descent update does not increase the function value.*

*Proof.* By Lemma 2,

$$f(x^{(k+1)}) \le f(x^{(k)}) + \langle \nabla f(x^{(k)}), x^{(k+1)} - x^{(k)} \rangle + \frac{L}{2} \|x^{(k+1)} - x^{(k)}\|_2^2$$

$$= f(x^{(k)}) - \eta \cdot \|\nabla f(x^{(k)})\|_2^2 + \frac{\eta^2 L}{2} \|\nabla f(x^{(k)})\|_2^2$$

$$= f(x^{(k)}) - \eta \cdot \left(1 - \frac{\eta L}{2}\right) \|\nabla f(x^{(k)})\|_2^2$$

$$\le f(x^{(k)}) - \frac{\eta}{2} \|\nabla f(x^{(k)})\|_2^2 \quad (\text{since } \eta L \le 1)$$

$$\le f(x^{(k)}) - \frac{1}{2L} \|\nabla f(x^{(k)})\|_2^2.$$

$$\square$$

Now we prove Theorem 1, which we restate below.

**Theorem 1.** *Let $\boldsymbol{A}_{i,j} \in \mathbb{R}^{p \times p}$ be a target block and $\boldsymbol{U}_i^{(k)}, \boldsymbol{V}_j^{(k)} \in \mathbb{R}^{p \times r}$, and $\boldsymbol{s}_{i,j}^{(k)} \in \mathbb{R}^r$ be factors of a block in the BLAST matrix to be optimized. With the step sizes $0 < \eta_{\boldsymbol{U}_i^{(k)}} \leq 1/\sigma_1\left(\bar{\boldsymbol{V}}_i^{(k)T} \bar{\boldsymbol{V}}_i^{(k)}\right)$, $0 < \eta_{\boldsymbol{V}_j^{(k)}} \leq 1/\sigma_1\left(\bar{\boldsymbol{U}}_j^{(k)T} \bar{\boldsymbol{U}}_j^{(k)}\right)$, $0 < \eta_{\boldsymbol{s}_{i,j}^{(k)}} \leq 1/\sigma_1\left((\boldsymbol{U}_i^{(k+1)T} \boldsymbol{U}_i^{(k+1)}) \odot (\boldsymbol{V}_j^{(k+1)T} \boldsymbol{V}_j^{(k+1)})\right)$, the gradient descent updates in Equations (5) to (7) monotonically non-increase the loss:*

$$\ell(\boldsymbol{U}_*^{(k+1)}, \boldsymbol{V}_*^{(k+1)}, \boldsymbol{s}_{*,*}^{(k+1)}) \leq \ell(\boldsymbol{U}_*^{(k)}, \boldsymbol{V}_*^{(k)}, \boldsymbol{s}_{*,*}^{(k)}).$$

*Proof.* To prove Theorem 1, we first show that each step of Equations (5) to (7) satisfies Lemma 3 under the given conditions. Then we resemble the results to construct the bound.

**Gradient Descent Update on $\boldsymbol{U}_i$** Let us denote the loss term regarding $\boldsymbol{U}_i$ by

$$\ell(\boldsymbol{U}_i) = \frac{1}{2} \left\| \boldsymbol{A}_{i,*} - \boldsymbol{U}_i \bar{\boldsymbol{V}}_i^T \right\|_F^2.$$

Since (i) the Frobenius norm is convex, (ii) all linear mappings are convex, and (iii) a composition of two convex functions are convex, $\left\| \boldsymbol{A}_{i,*} - \boldsymbol{U}_i \bar{\boldsymbol{V}}_i^T \right\|_F^2$ is a convex function of $\boldsymbol{U}_i$. Also, the gradient we derived in Equation (10) always exists and has the following vectorized form:

$$\begin{aligned}
\text{vec}(\nabla\ell(\boldsymbol{U}_i)) &= \text{vec}(\boldsymbol{U}_i \bar{\boldsymbol{V}}_i^T \bar{\boldsymbol{V}}_i) - \text{vec}(\boldsymbol{A}_{i,*} \bar{\boldsymbol{V}}_i) \\
&= ((\bar{\boldsymbol{V}}_i^T \bar{\boldsymbol{V}}_i)^T \otimes \boldsymbol{I})\boldsymbol{u}_i - \text{vec}(\boldsymbol{A}_{i,*} \bar{\boldsymbol{V}}_i),
\end{aligned}$$

where $\otimes$ denotes the Kronecker product and $\boldsymbol{u}_i = \text{vec}(\boldsymbol{U}_i)$. The Lipschitz constant of the gradient is the largest singular value of the matrix $(\bar{\boldsymbol{V}}_i^T \bar{\boldsymbol{V}}_i)^T \otimes \boldsymbol{I}$, which is the largest singular value of $\bar{\boldsymbol{V}}_i^T \bar{\boldsymbol{V}}_i$ since the Kronecker product of two matrices of singular values $\boldsymbol{\Sigma}_1$ and $\boldsymbol{\Sigma}_2$ has the singular values of $\boldsymbol{\Sigma}_1 \otimes \boldsymbol{\Sigma}_2$ (see [62, Theorem 4.2.15]).

Therefore, from Lemma 3, we obtain the following bound:

$$\ell(\boldsymbol{U}_i^{(k+1)}) \leq \ell(\boldsymbol{U}_i^{(k)}) - \frac{\|\nabla_{\boldsymbol{U}_i^{(k)}} \ell(\boldsymbol{U}_i^{(k)})\|_F^2}{2\sigma_1\left(\bar{\boldsymbol{V}}_i^{(k)T} \bar{\boldsymbol{V}}_i^{(k)}\right)}, \quad i = 1, \ldots, b. \tag{23}$$

**Gradient Descent Update on $\boldsymbol{V}_j$** The loss function $\ell(\boldsymbol{V}_j) = \frac{1}{2}\|\boldsymbol{A}_{*,j} - \bar{\boldsymbol{U}}_j \bar{\boldsymbol{U}}_j^T\|_F^2$ with respect to $\boldsymbol{V}_j$ is convex to $\boldsymbol{V}_j$ and the gradient of $\boldsymbol{V}_j$ in Equation (11) also always exists and can be rewritten as follows:

$$\begin{aligned}
\text{vec}(\nabla\ell(\boldsymbol{V}_j)) &= \text{vec}(\boldsymbol{V}_j \bar{\boldsymbol{U}}_j^T \bar{\boldsymbol{U}}_j) - \text{vec}(\boldsymbol{A}_{*,j}^T \bar{\boldsymbol{U}}_j) \\
&= ((\bar{\boldsymbol{U}}_j^T \bar{\boldsymbol{U}}_j) \otimes \boldsymbol{I})\boldsymbol{v}_j - \text{vec}(\boldsymbol{A}_{*,j}^T \bar{\boldsymbol{U}}_j).
\end{aligned}$$

The Lipschitz constant of the gradient is again the largest singular value of $\bar{\boldsymbol{U}}_j^T \bar{\boldsymbol{U}}_j$. We have the bound from Lemma 3 similar to Equation (23):

$$\ell(\boldsymbol{V}_j^{(k+1)}) \leq \ell(\boldsymbol{V}_j^{(k)}) - \frac{\|\nabla_{\boldsymbol{V}_j^{(k)}} \ell(\boldsymbol{V}_i^{(k)})\|_F^2}{2\sigma_1\left(\bar{\boldsymbol{U}}_j^{(k)T} \bar{\boldsymbol{U}}_j^{(k)}\right)}, \quad j = 1, \ldots, b. \tag{24}$$

**Gradient Descent Update on $\boldsymbol{s}_{i,j}$** The loss function

$$\ell(\boldsymbol{s}_{i,j}) = \frac{1}{2} \left\| \boldsymbol{A}_{i,j} - \boldsymbol{U}_i \text{diag}(\boldsymbol{s}_{i,j}) \boldsymbol{V}_j^T \right\|_F^2$$

is also convex in $\boldsymbol{s}_{i,j}$ since $\text{diag}(\cdot)$ is a convex mapping. We know the gradient exists from Equation (15):

$$\begin{aligned}
\nabla\ell(\boldsymbol{s}_{i,j}) &= \text{diag}\left(\boldsymbol{U}_i^T \boldsymbol{U}_i \text{diag}(\boldsymbol{s}_{i,j}) \boldsymbol{V}_j^T \boldsymbol{V}_j\right) - \text{diag}\left(\boldsymbol{U}_i^T \boldsymbol{A}_{i,j} \boldsymbol{V}_j\right) \\
&= \left((\boldsymbol{U}_i^T \boldsymbol{U}_i) \odot (\boldsymbol{V}_j^T \boldsymbol{V}_j)\right) \boldsymbol{s}_{i,j} - \text{diag}\left(\boldsymbol{U}_i^T \boldsymbol{A}_{i,j} \boldsymbol{V}_j\right),
\end{aligned}$$

and the Lipschitz constant of the gradient is $\sigma_1\left(\left(\boldsymbol{U}_i^T\boldsymbol{U}_i\right) \odot \left(\boldsymbol{V}_j^T\boldsymbol{V}_j\right)\right)$. The bound from Lemma 3 for the diagonal factors is as follows:

$$\ell(\boldsymbol{s}_{i,j}^{(k+1)}) \leq \ell(\boldsymbol{s}_{i,j}^{(k)}) - \frac{\|\nabla_{\boldsymbol{s}_{i,j}^{(k)}}\ell(\boldsymbol{s}_{i,j}^{(k)})\|_2^2}{2\sigma_1\left(\left(\boldsymbol{U}_i^T\boldsymbol{U}_i\right) \odot \left(\boldsymbol{V}_j^T\boldsymbol{V}_j\right)\right)}, \quad i,j = 1,\ldots,b. \tag{25}$$

Combining Equations (23) to (25), we retrieve the bound in Theorem 1:

$$\ell(\boldsymbol{U}_*^{(k+1)}, \boldsymbol{V}_*^{(k)}, \boldsymbol{s}_{*,*}^{(k)}) \leq \ell(\boldsymbol{U}_*^{(k)}, \boldsymbol{V}_*^{(k)}, \boldsymbol{s}_{*,*}^{(k)}),$$
$$\ell(\boldsymbol{U}_*^{(k+1)}, \boldsymbol{V}_*^{(k+1)}, \boldsymbol{s}_{*,*}^{(k)}) \leq \ell(\boldsymbol{U}_*^{(k+1)}, \boldsymbol{V}_*^{(k)}, \boldsymbol{s}_{*,*}^{(k)}),$$
$$\ell(\boldsymbol{U}_*^{(k+1)}, \boldsymbol{V}_*^{(k+1)}, \boldsymbol{s}_{*,*}^{(k+1)}) \leq \ell(\boldsymbol{U}_*^{(k+1)}, \boldsymbol{V}_*^{(k+1)}, \boldsymbol{s}_{*,*}^{(k)})$$
$$\implies \ell(\boldsymbol{U}_*^{(k+1)}, \boldsymbol{V}_*^{(k+1)}, \boldsymbol{s}_{*,*}^{(k+1)}) \leq \ell(\boldsymbol{U}_*^{(k)}, \boldsymbol{V}_*^{(k)}, \boldsymbol{s}_{*,*}^{(k)})$$

$\square$

# C   Experimental Details

In this section, we provide the experimental details. Throughout the experiments, we used 8 NVIDIA A40 GPUs or 4 NVIDIA L40S GPUs for training and evaluation, and a single NVIDIA A100 GPU with 40GB memory for runtime evaluation.

## C.1   Datasets and Benchmarks

**Image Datasets**   For image classification tasks, we use CIFAR-10 [28], CIFAR-100 [28], and ImageNet-1k [29] datasets for our experiments. CIFAR-10 and 100 contain 50,000 training and 10,000 test images, each of which is $32 \times 32$ color images of 10 and 100 classes, respectively. ImageNet-1k consists of 1,281,167 training and 50,000 validation images of 1,000 classes.

**Common Sense Reasoning Benchmarks**   For our large language model evaluation, we use the following common sense reasoning benchmarks: Physical Interaction: Question Answering (PIQA) [35], HellaSwag[36], WinoGrande[37], BoolQ[38], OpenBookQA[39], AI2's Reasoning Challenge (ARC)-easy and challenge [40]. PIQA targets the task of physical common sense reasoning, with 16,000 examples for training, 2,000 for development, and 3,000 for testing. HellaSwag is composed of 10k questions that are specifically hard for the machines, though trivial for humans (95% accuracy). Winogrande is a large-scale dataset of 44k pronoun resolution problems. BoolQ consists of 15,942 yes/no questions. OpenBookQA is modeled after open book exams for assessing human understanding of a subject, containing 5,957 multiple-choice elementary-level science questions (4,957 train, 500 dev, 500 test). AI2's Reasoning Challenge (ARC) dataset consists of 7,787 multiple-choice science exam questions, and the questions are categorized into "easy" and "challenging" subsets.

## C.2   CIFAR-10/100 and ImageNet-1k Image Classification Training

For CIFAR-10 and CIFAR-100 training, we trained the ViT-Small models with $4 \times 4$-sized patches [19]. We trained ViT-Base models with $16 \times 16$-sized patches for ImageNet-1k training. All models were trained by the AdamW [22] optimizer.

In the ViT models, we replaced the weight matrices of query, key, and value projection layers in one attention module and those in the feed-forward modules. In addition, we stacked the weights of query, key, and value weights and modeled them by one BLAST matrix.

The BLAST factors were randomly initialized to have the desired standard deviation $\sigma = 0.02$ while having zero-mean, where the standard deviation $0.02$ was also used for initializing the weights of the original-sized ViTs. Specifically, we initialized the factors as follows:

$$\boldsymbol{U}_i \sim \mathcal{N}(\boldsymbol{0}, \sqrt{0.02}\boldsymbol{I}), \quad \forall i = 1, 2, \ldots, b,$$
$$\boldsymbol{V}_j \sim \mathcal{N}(\boldsymbol{0}, \sqrt{0.02}\boldsymbol{I}), \quad \forall j = 1, 2, \ldots, b,$$
$$\boldsymbol{s}_{i,j} \sim \text{Unif}(0.0, 2.0), \quad \forall i, j = 1, 2, \ldots, b,$$

where $\mathrm{Unif}(0.0, 2.0)$ denotes a uniform distribution on the interval $[0, 2]$.

For all models, we applied AutoAugment [63]. We summarize the training hyperparameters in Table 5.

| Dataset | Model | Epochs | Weight Decay | Batch Size | Warmup Epochs | Warmup Start | LR Scheduler | LR | LR Min | Dropout | Droppath | BLAST $b$ |
|---------|-------|--------|--------------|------------|---------------|--------------|--------------|-----|--------|---------|----------|-----------|
| CIFAR-10 CIFAR-100 | ViT-S | 310 | 0.05 | 1024 | 5 | 1e-6 | cosine | 5e-4 | 1e-5 | 0 | 0.1 | 3 |
| ImageNet | ViT-B | 310 | 0.05 | 1024 | 10 | 1e-6 | cosine | 1e-3 | 1e-5 | 0 | 0 | 3 |

Table 5: Hyperparameters used in training from scratch.

### C.3 Compression and Re-training

**ViT on ImageNet-1k**   For ImageNet-1k compression and re-training, we followed a similar strategy to the ImageNet training with minor changes, summarized in Table 6. The ViT-Base with $16 \times 16$-sized patches was chosen as a baseline model. We decomposed the pre-trained weight matrices of ViT-Base by Algorithm 2 with $K = 300$ and $\delta_0 = 0.1$. Also, we linearly decayed the step size from 1.0 to 0.0. Then, all compressed models were trained by the AdamW [22] optimizer.

| Dataset | Model | Epochs | Steps | Weight Decay | Batch Size | Warmup Steps | Warmup Start | LR Scheduler | LR | LR Min | Dropout | Droppath | BLAST $b$ |
|---------|-------|--------|-------|--------------|------------|--------------|--------------|--------------|-----|--------|---------|----------|-----------|
| ImageNet | ViT-B | 35 | - | 0.05 | 1024 | 0 | N/A | cosine | 2e-4 | 1e-5 | 0 | 0.1 | 3 or 12 |
| SlimPajama | Llama-7B | - | 400 | 0.0 | 144 | 12 | 1.67e-5 | cosine | 2e-4 | 0 | 0 | 0 | 16 |

Table 6: Hyperparameters used in re-training.

**Diffusion Model**   We compressed the DiT-XL model with $2 \times 2$-sized patches [13], pre-trained on the $256 \times 256$ ImageNet training samples. A DiT model is a variant of Vision Transformer [19] with additional adaptive layer normalization (adaLN) [64]. Here, we compressed the stacked query, key, and value weights, the first fully connected layer of the feed-forward module, and the adaLN projection layers by $\mathrm{BLAST}_9$ or low-rank matrices. All weights were compressed by Algorithm 2 with $K = 500$ and $\delta_0 = 0.1$. We linearly decayed the step size from 1.0 to 0.0. The parameters of the BLAST and the low-rank matrices were set to have the desired compression ratio in total, i.e., we remove 50% (or 20%) of the total parameters out of the network. We present the summary in Tables 7 and 8. We generated 50,000 images using the original, the low-rank-compressed, and the BLAST-compressed DiT.

| Layer Type | $m$ | $n$ | $b$ | $r$ | Layer Indices |
|------------|-----|-----|-----|-----|---------------|
| QKV_proj | 3456 | 1152 | 9 | 384 | 0-27 |
| FC1 | 4608 | 1152 | 9 | 256 | 0-27 |
| adaLN_proj | 6912 | 1152 | 9 | 256 | 0-27 |

Table 7: Hyperparameters used for the $\mathrm{BLAST}_9$-compressed DiT-XL/2 with 50% compression ratio. $m, n$: size of the original matrix, $b$: number of row/column partitions, $r$: BLAST rank parameter, Layer Indices: indices of layers that the BLAST matrix replaces the weight.

**FID, sFID and IS Evaluation**   We sampled the novel images using DDPM sampler [65] for FID, sFID, and IS evaluation in Table 2. The step size was set to 250 for each model. Then, the FID, sFID, and IS were computed between each pool of generated images and the 50,000 ImageNet validation images to estimate the distributional discrepancy between the target and the generated samples.

**Large Language Model**   For the large language model compression, we used the Llama-7B pre-trained model, publicly available at https://huggingface.co/huggyllama/llama-7b. For the 50% compression ratio, we compressed all the weights in the main modules of Llama-7B with BLAST with the parameters described in Table 9. For the 20% and 10% compression ratios, we compressed the weights of Q_proj, K_proj, gate_proj, up_proj layers to match the target compression ratio of the total parameter counts. Also, by following [66], we compress Q_proj and K_proj layers for the first 10 attention modules. The parameters used in the experiment are summarized in Tables 8, 9 and 11. All weights were compressed by Algorithm 2 with $K = 300$ and

| Layer Type | $m$ | $n$ | $b$ | $r$ | Layer Indices |
|---|---|---|---|---|---|
| QKV_proj | 3456 | 1152 | 9 | 512 | 0-27 |
| FC1 | 4608 | 1152 | 9 | 640 | 0-27 |
| adaLN_proj | 6912 | 1152 | 9 | 768 | 0-27 |

Table 8: Hyperparameters used for the BLAST$_9$-compressed DiT-XL/2 with 20% compression ratio. $m, n$: size of the original matrix, $b$: number of row/column partitions, $r$: BLAST rank parameter, Layer Indices: indices of layers that the BLAST matrix replaces the weight.

$\delta_0 = 0.1$. We linearly decayed the step size from 1.0 to 0.0. The factorization process takes 3.38 GPU hours for the BLAST weights of $b = 16$ on NVIDIA A40 GPUs.

To re-train the compressed Llama models, we used a subset[2] of the SlimPajama dataset [34] for 400 steps using 0.47B tokens. The global batch size was set to 576, and the models were trained on 4 NVIDIA L40S GPUs. See Table 6 for details.

We used Language Model Evaluation Harness[3] [67] for the zero-shot classification accuracy evaluation.

| Layer Type | $m$ | $n$ | $b$ | $r$ | Layer Indices |
|---|---|---|---|---|---|
| Q_proj | 4096 | 4096 | 16 | 1024 | 0-31 |
| K_proj | 4096 | 4096 | 16 | 1024 | 0-31 |
| V_proj | 4096 | 4096 | 16 | 1024 | 0-31 |
| O_proj | 4096 | 4096 | 16 | 1024 | 0-31 |
| gate_proj | 11008 | 4096 | 16 | 1488 | 0-31 |
| up_proj | 11008 | 4096 | 16 | 1488 | 0-31 |
| down_proj | 4096 | 11008 | 16 | 1488 | 0-31 |

Table 9: Hyperparameters used for the BLAST$_{16}$-compressed Llama-7B with 50% compression ratio. $m, n$: size of the original matrix, $b$: number of row/column partitions, $r$: BLAST rank parameter, Layer Indices: indices of layers that the BLAST matrix replaces the weight.

| Layer Type | $m$ | $n$ | $b$ | $r$ | Layer Indices |
|---|---|---|---|---|---|
| Q_proj | 4096 | 4096 | 16 | 496 | 0-31 |
| K_proj | 4096 | 4096 | 16 | 496 | 0-31 |
| gate_proj | 11008 | 4096 | 16 | 2048 | 10-31 |
| up_proj | 11008 | 4096 | 16 | 2048 | 10-31 |

Table 10: Hyperparameters used for the BLAST$_{16}$-compressed Llama-7B with 20% compression ratio. $m, n$: size of the original matrix, $b$: number of row/column partitions, $r$: BLAST rank parameter, Layer Indices: indices of layers that the BLAST matrix replaces the weight.

## D  Additional Experimental Results

### D.1  Synthetic Experiments on BLAST Factorization

In this experiment, we test the factorization algorithms discussed in Section 3, similar to Figure 3 but with a different target matrix. To be specific, we synthesize a $256 \times 256$-sized BLAST$_{16}$ (i.e., $b = 16$) target matrix with $r^* = 8$. Then, we compare the error curve along the iterates of the gradient descent without preconditioning (GD) in Equations (5) to (7), and the preconditioned gradient descent (PrecGD) in Algorithm 2. Here, we consider the exact parameterization setting when $r = r^* = 8$ and the over-parameterized setting $r = 32 > r^*$. In Figure 9, unlike the low-rank target matrix, GD does not converge in both cases. However, the preconditioned version easily finds the low-error solution for the exact parameterization. For the overparameterized case, the preconditioned gradient descent method in Algorithm 2 achieved in two orders of magnitude improvement from the simple GD.

---

[2]https://huggingface.co/datasets/DKYoon/SlimPajama-6B
[3]https://github.com/EleutherAI/lm-evaluation-harness

| Layer Type | $m$ | $n$ | $b$ | $r$ | Layer Indices |
|---|---|---|---|---|---|
| Q_proj | 4096 | 4096 | 16 | 1024 | 0-31 |
| K_proj | 4096 | 4096 | 16 | 1024 | 0-31 |
| gate_proj | 11008 | 4096 | 16 | 2368 | 10-31 |

Table 11: Hyperparameters used for the BLAST$_{16}$-compressed Llama-7B with 10% compression ratio. $m, n$: size of the original matrix, $b$: number of row/column partitions, $r$: BLAST rank parameter, Layer Indices: indices of layers that the BLAST matrix replaces the weight.

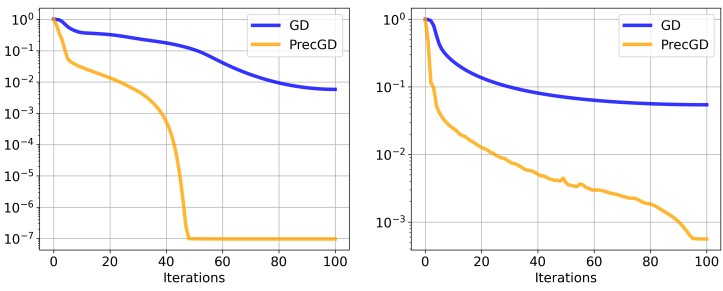

BLAST $\rightarrow$ BLAST.

Figure 9: Plots of normalized reconstruction errors using the BLAST factorization with GD and GD with preconditioning steps (PrecGD) in both exact and rank overparameterized settings, when the target matrix is BLAST$_{16}$. Left: Reconstruction errors when $r = r^*$. Right: Reconstruction errors when $r > r*$.

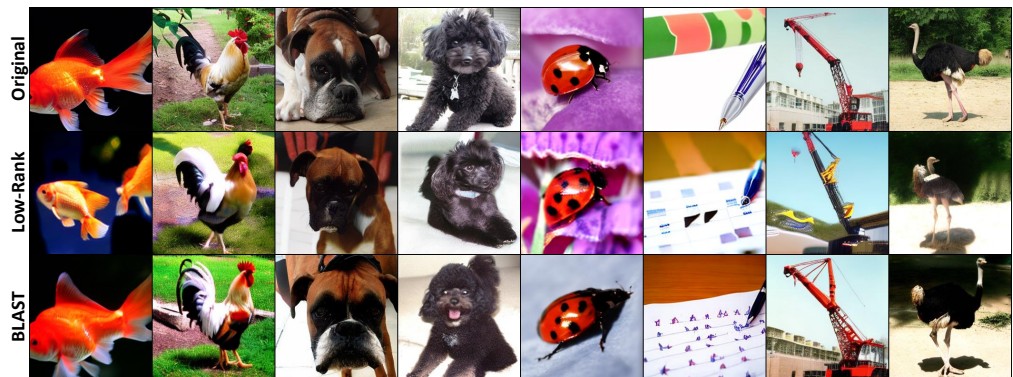

Figure 10: Examples of generated images using both low-rank and BLAST decompositions. Both methods compress the original model by 20%.

## D.2 Additional Results on Diffusion Model Compression

We include extended experimental results of the diffusion model compression in Section 4.2

**Compression-only** In Figures 10 and 11, the additional image samples of original uncompressed, low-rank-compressed, and BLAST$_9$-compressed DiT [13] models are presented. The images in the same column were sampled using 250 DDIM steps, starting from the same noise vector. The compression ratio was set to 20% for both models. The figures show that the outputs of the model compressed by BLAST maintain similar features and perceptual quality to the outputs of the original DiT.

**Compression and re-training** We present additional samples from the low-rank and BLAST DiT models at the 50% compression ratio after *re-training* in Figure 13. Similar to Figure 1, the images generated by the low-rank DiT lose significant image quality, whereas the images from the BLAST DiT preserve the quality and semantics of the original samples.

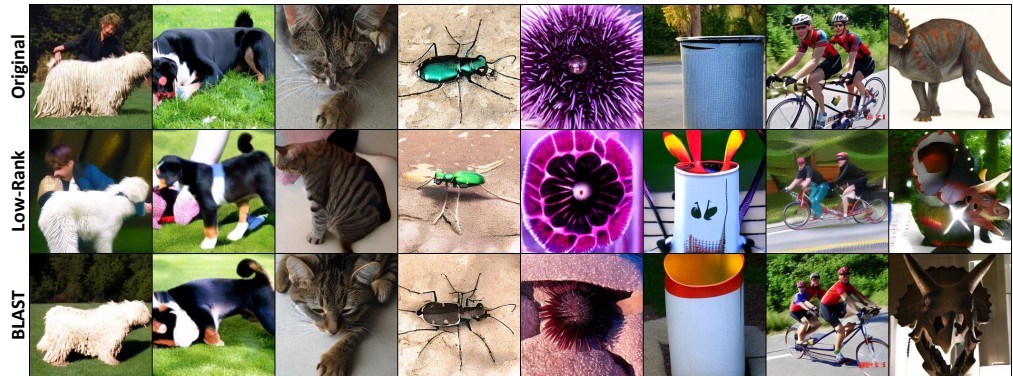

Figure 11: More examples of generated images using both low-rank and BLAST decompositions. Both methods compress the original model by 20%.

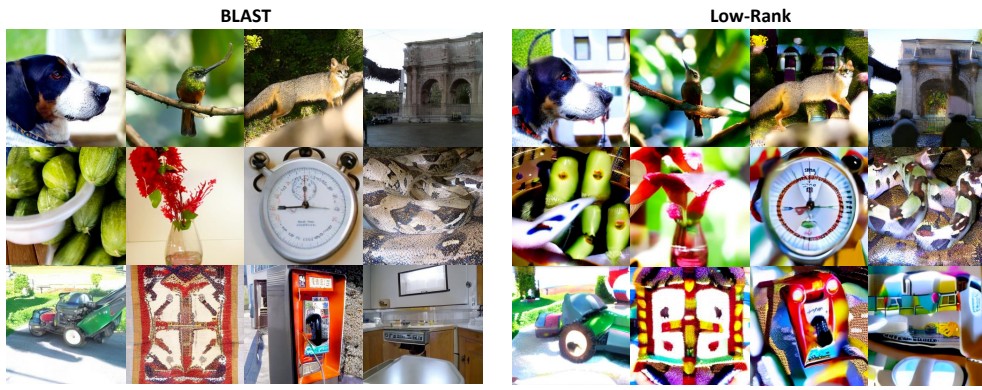

Figure 12: Comparison of the images generated by the BLAST and low-rank compressed models. Overall, the low-rank approximated model often generates unrealistic images, which contributes to low scores evident in Table 2.

**Evidence of low performance of low-rank-compressed model**   Some images generated by the 20% low-rank-compressed model (Figure 12-left) are highly unrealistic and have inferior quality compared to the images generated by original and BLAST-compressed models (Figure 12-right). We observe that these samples contribute to the low scores in Table 2. These samples were computed using 250 DDPM steps, as done by the original DiT work [13].

### D.3   Additional Results on Large Language Model Compression

**Compression-only**   We report the performance of LLM-Pruner [47] and Joint Rank-$k$ [66] for the same compression task. LLM-Pruner [47] identifies sparse weights *with* data to pinpoint unimportant neurons. Joint Rank-$k$ [66] performs a low-rank approximation *jointly* on weight matrices with similar column spaces by stacking them and applying a truncated SVD.

In Table 12, we compare the performance degradation of LLM-Pruner, Joint Rank-$k$, Low-Rank, $BLAST_2$, and $BLAST_{16}$, as well as their absolute performance. The first four rows represent the zero-shot performance of Llama-7B [1] from the literature, while the row marked with an asterisk (*) indicates our results.

For 10% compression, Joint Rank-$k$ achieved the lowest performance degradation, although $BLAST_{16}$ also exhibited a similar performance drop. When the model is compressed by 20%, $BLAST_{16}$ surpasses Joint Rank-$k$ [66], LLM-Pruner [47], and low-rank schemes. The zero-shot accuracy versus compression ratio curve in Figure 7 shows that BLAST compression results in less performance drop for the same compression ratio.

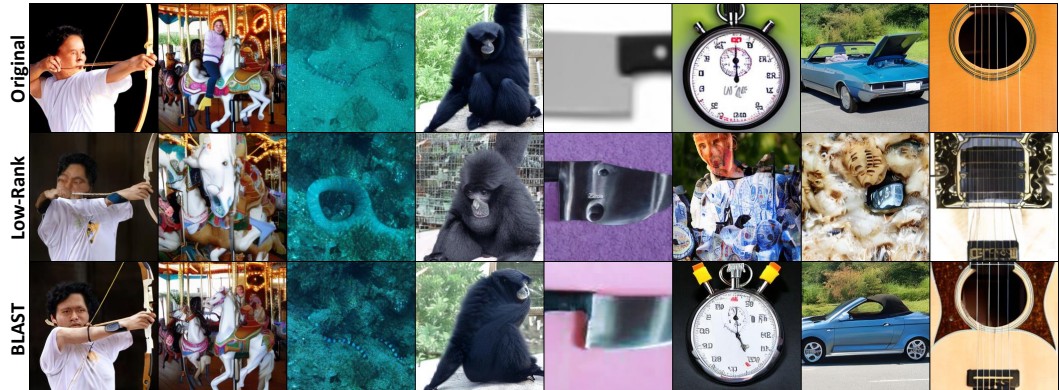

Figure 13: Examples of generated images using DiT [13] starting from the same noise vectors and a deterministic solver. The original model is compressed by 50% through BLAST or Low-Rank matrices and re-trained for 10 epochs on ImageNet. The images from the model compressed via BLAST preserves the quality of the images of the original model, whereas the images generated by the low-rank model contain artifacts.

| CR | Method | PIQA | HellaSwag | Winogrande | BoolQ | OBQA | ARC-e | ARC-c | Average |
|---|---|---|---|---|---|---|---|---|---|
| 0% | LLaMA-7B[1] | 79.8 | 76.1 | 70.1 | 76.5 | 57.2 | 72.8 | 47.6 | 68.59 |
| | LLaMA-7B[47] | 78.35 | 72.99 | 67.01 | 73.18 | 42.40 | 67.45 | 41.38 | 63.25 |
| | LLaMA-7B[66] | 77.64 | 73.08 | 62.12 | 69.33 | 43.40 | 66.31 | 37.63 | 61.36 |
| | LLaMA-7B* | 79.16 | 76.19 | 70.09 | 75.11 | 44.4 | 72.9 | 44.71 | 66.08 |
| 10% | Joint Rank-$k$[66] | 76.93(-0.71) | 71.67(-1.41) | 62.27(+0.15) | 67.58(-1.75) | 43.00(-0.40) | 66.49(+0.18) | 36.61(-1.02) | 60.62(-0.74) |
| | Joint Rank-$k$*[66] | 77.91(-1.25) | 71.78(-4.41) | 68.98(-1.11) | 74.01(-1.10) | **44.80(+0.40)** | **72.43(-0.47)** | 41.72(-2.99) | 64.52(-1.56) |
| | Monarch*[14] | 77.37(-1.79) | 69.10(-7.09) | 67.96(-2.13) | 71.71(-3.40) | 43.60(-0.80) | 69.57(-3.33) | 40.61(-4.10) | 62.85(-3.23) |
| | BLAST$_{16}$ | **78.78(-0.38)** | **74.22(-1.97)** | **70.24(+0.15)** | **76.12(+1.01)** | 42.00(-2.40) | 71.21(-1.69) | **43.26(-1.45)** | **65.12(-0.96)** |
| 20% | LLM-Pruner[47] | 75.68(-2.67) | 66.80(-6.19) | 59.83(-7.18) | 57.06(-16.12) | 40.00(-2.40) | 60.94(-6.51) | 36.52(-4.86) | 56.69(-6.56) |
| | Joint Rank-$k$[66] | 75.08(-2.56) | 64.57(-8.51) | 60.46(-1.66) | 62.20(-7.13) | 43.00(-0.40) | 61.73(-4.58) | 34.24(-3.39) | 57.33(-4.03) |
| | Joint Rank-$k$*[66] | 75.90(-3.26) | 65.53(-10.66) | 66.85(-3.24) | 67.58(-7.53) | 42.20(-2.20) | 66.79(-6.11) | 38.65(-6.06) | 60.50(-5.58) |
| | Low-Rank* | 75.30(-3.86) | 63.20(-12.99) | 65.11(-4.98) | 66.64(-8.47) | 42.20(-2.20) | 65.91(-6.99) | 38.65(-6.06) | 59.57(-6.51) |
| | Monarch*[14] | 72.31(-6.85) | 42.38(-33.81) | 54.85(-15.24) | 62.20(-12.91) | 31.40(-13.40) | 51.47(-21.43) | 27.73(-16.98) | 48.91(-17.17) |
| | BLAST$_2$ | 76.12(-3.04) | 66.29(-9.90) | 65.19(-4.90) | 72.17(-2.94) | 43.60(-0.80) | 67.26(-5.64) | 40.19(-4.52) | 61.55(-4.53) |
| | BLAST$_{16}$ | **77.48(-1.68)** | **69.74(-6.45)** | **68.03(-2.06)** | **72.45(-2.66)** | **44.00(-0.40)** | **68.64(-4.26)** | **40.27(-4.44)** | **62.94(-3.14)** |

Table 12: Zero-shot performance of LLaMA-7B with various compression methods *without* retraining. All models are *not* post-trained. CR denotes compression ratio. **Bold** indicates the best performance under the same compression ratio. Underline refers to the lowest performance drop. BLAST$_b$ indicates the BLAST matrix with $b \times b$ number of blocks. The mark * represents the results from our experiment.

**Compression and Re-training**   In Table 13, we present the performance of each common sense reasoning benchmark which we report their average in Table 3.

| CR | Method | PIQA | HellaSwag | Winogrande | BoolQ | OBQA | ARC-e | ARC-c | Average |
|---|---|---|---|---|---|---|---|---|---|
| 0% | LLaMA-7B | 79.16 | 76.23 | 69.93 | 75.14 | 44.4 | 72.9 | 44.71 | 66.07 |
| 20% | BLAST$_{16}$ | 77.97(-1.19) | 72.88(-3.35) | 69.30(-0.63) | 72.63(-2.51) | 41.20(-3.20) | 70.33(-2.57) | 41.81(-2.90) | 63.73(-2.34) |
| 50% | Low-Rank | 66.16(-13.00) | 48.31(-27.92) | 54.78(-15.15) | 65.38(-9.76) | 31.00(-13.40) | 45.41(-27.49) | 27.73(-16.98) | 48.40(-17.67) |
| | Monarch | 50.71(-28.45) | 26.17(-50.06) | 49.33(-20.60) | 37.86(-37.28) | 26.40(-18.00) | 26.64(-46.26) | 28.07(-16.64) | 35.03(-31.04) |
| | Block-Diagonal | 50.38(-28.78) | 26.24(-49.99) | 50.59(-19.34) | 37.83(-37.31) | 24.80(-19.60) | 26.35(-46.55) | 27.82(-16.89) | 34.86(-31.21) |
| | BLAST$_{16}$ | 73.83(-5.33) | 63.59(-12.64) | 63.38(-6.55) | 68.62(-6.52) | 34.60(-9.80) | 57.24(-15.66) | 32.34(-12.37) | 56.23(-9.84) |

Table 13: Zero-shot performance of LLaMA-7B with various compression methods *after* re-training. CR denotes compression ratio. BLAST$_b$ indicates the BLAST matrix with $b \times b$ number of blocks.

# E   Broader Impact

Our proposed method targets improving the efficiency of the DNN inference. This might have a negative social impact by promoting the accessibility and usability of malicious DNNs such as DeepFake. However, at the same time, we expect the BLAST matrix will bring a tremendous positive social impact. First, it contributes to sustainability by cutting down the energy consumption for the DNN inference. Moreover, the BLAST matrix can improve the accessibility of AI-based medical, educational, and social services by providing a foundation for running those models on mobile devices. Therefore, we believe BLAST will give tangible benefits to our society.

