# OpenReview forum: "BLAST: Block-Level Adaptive Structured Matrices for Efficient Deep Neural Network Inference"
_NeurIPS.cc/2024/Conference — NeurIPS 2024 poster_

### Official Review · Reviewer_2dKz · 2024-07-12

**Soundness:** 3
**Presentation:** 2
**Contribution:** 2
**Rating:** 4
**Confidence:** 4

**Summary:**

This paper proposes to compress linear layers by using a block low-rank structure with shared basis -- also known as the BLR^2 matrix structure. The basis U, V, and diagonal weights s of the low-rank block are computed through a gradient-based optimization that minimizes the Frobenius norm between the dense and low-rank matrix. They also use preconditioning to accelerate the convergence.

**Strengths:**

Their experiments of ViT-s on ImageNet-1k show an improvement in validation accuracy compared to existing methods such as Gaudi-GBLR.

**Weaknesses:**

What is proposed as the BLAST matrix in this paper is an existing matrix structure known as BLR^2 [https://hal.science/hal-03070416]. The original BLR^2 structure uses LQ decomposition to compress the low-rank blocks, but the name of the structure is independent of the method used to compress the low-rank blocks. Therefore, the proposed method which uses a gradient based iterative method to compress the low-tank blocks is still a BLR^2 structure. The BLR^2 paper should be cited, and all mention of BLAST should be replaced with BLR^2.

For CIFAR-10 and CIFAR-100, Gaudi-GBLR seems to perform better than the proposed method. The authors mention that Gaudi-GBLR's "capability of learning the adaptive resource/budget allocation for each weight matrix" is the cause, but it is unclear why this is not the case for ImageNet-1k.

**Questions:**

Why does the paper repeatedly mention matrix-vector multiplications? The input to DNNs are not vectors.

Why do the authors mention that "A block low-rank (BLR) [12], also known as Monarch [13] matrix"? The Monarch matrix is a very different structure from BLR.

Why are Gaudi-GBLR, BSP+LR, and BLR excluded from the compression benchmark in Table 2? Just because they haven't been studied as data-free compression techniques doesn't mean they are not suitable for this task.

**Limitations:**

The limitations are stated clearly in a separate section.

---

> ### Author Rebuttal · Authors · 2024-08-06
>
> ### Q1. The BLAST matrix in this paper is an existing matrix structure known as BLR^2. The BLR^2 paper should be cited, and all mention of BLAST should be replaced with BLR^2.
>
> Thank you for suggesting an important related work\! We agree that BLR^2 \[Ashcraft, Buttari & Mary, 2021\] should be cited and carefully discussed.
>
> However, after carefully reviewing both approaches, we concluded that BLAST is **not identical to** BLR^2. One *cannot* replace BLAST with BLR^2 to obtain the same level of generality and efficiency that BLAST provides. Therefore, we think BLAST should be named as-is.
>
> The BLAST matrix structure differs from the BLR^2 \[Ashcraft, Buttari & Mary, 2021\] method in its left, right, and diagonal factors.
> (i) The matrix ($S\_{i,j}$) of a BLAST block ($A\_{i,j}=U\_i S\_{i,j} V\_j^T$) is a diagonal matrix, whereas BLR^2 uses a *low-rank* matrix. Accordingly, a BLAST matrix can capture **high-rank blocks** with **fewer parameters** than BLR^2. (ii) BLR^2 restricts its shared left and right factors to have *orthonormal* bases, which introduces extra overhead and instability at each gradient descent update. In contrast, *none* of the BLAST factors have such orthonormal constraints so that the training process does not change from the conventional DNN training scheme. In revision, we will cite BLR^2 and discuss the work.
>
> We argue that  BLAST is more suitable for efficient DNN inference and simple training than BLR^2 for the following reasons:
>
> **Expressiveness:** BLR^2 uses a low-rank coupling matrix whereas BLAST utilizes a diagonal matrix. As a consequence, BLAST can represent matrices that BLR^2 cannot. BLR^2's low-rank coupling factor fails to capture high-rank blocks, whereas BLAST's diagonal coupling factor can model high-rank blocks with fewer parameters.
> **Efficiency:** Each low-rank block of a BLR^2 matrix can have a *different* rank, making it challenging to achieve optimal efficiency on the off-the-shelf GPUs due to *zero paddings*. On the other hand, the diagonal factor of a BLAST matrix *does not require zero padding*, allowing for faster matrix multiplication on GPUs, as illustrated in Algorithm 1 and the pseudocode in "General Comments."
> **Gradient Descent Compatibility:** BLR^2's orthonormality constraint adds computational cost and instability during gradient descent steps in training and fine-tuning. In contrast, BLAST avoids these issues as it does not require modifications to the training process.
>
> Therefore, unlike BLR^2, BLAST exhibits unique properties beneficial for efficient DNN inference as claimed in our paper. Once again, we truly appreciate Reviewer 2dKz's constructive suggestion to discuss BLR^2 in our work.
>
> ### Q2. For CIFAR-10 and CIFAR-100, Gaudi-GBLR seems to perform better than the proposed method.
>
> Please refer to Q3 of “Global Comments.”
>
> ### Q3. Why does the paper repeatedly mention matrix-vector multiplications? The input to DNNs are not vectors.
>
> Since matrix-matrix multiplication is composed of multiple parallelized matrix-vector multiplication, we discussed BLAST at the matrix-vector multiplication level.
> We also thought the vector-shaped inputs were easier to express than the matrix-shaped or tensor-shaped inputs.
> Please refer to Q4 of “Global Comments” for the actual implementation.
>
> ### Q4. Why do the authors mention that "A block low-rank (BLR) \[12\], also known as Monarch \[13\] matrix"? The Monarch matrix is a very different structure from BLR.
>
> Thank you for pointing it out. We assumed the BLR matrix has the blocks of the same rank. In this case, BLR and Monarch are *equivalent* up to the row permutations. Indeed, the Monarch matrix is decomposed by the block-level SVD, which results in the block low-rank matrix of the same block rank. We will clarify that we assume the same block rank for BLR.
>
> ### Q5. Why are Gaudi-GBLR, BSP+LR, and BLR excluded from the compression benchmark in Table 2?
>
> Please refer to Q2 of “Global Comments.”
>
> We truly appreciate Reviewer 2dKz’s feedback. Please let us know whether our comments have successfully resolved all of your concerns or not. If so, we kindly request reconsidering the evaluation of our work.
>
> **Reference**
>
> Ashcraft C, Buttari A, Mary T. Block Low-Rank Matrices with Shared Bases: Potential and Limitations of the BLR ^2 Format. SIAM Journal on Matrix Analysis and Applications. 2021;42(2):990-1010.

---

> > ### Comment · Reviewer_2dKz · 2024-08-13
> >
> > Q1. I acknowledge the difference between BLR^2 and BLAST. I still believe that the terminology to describe the matrix structure should be decoupled from the methods used to compress them. For example, the blocks of the BLR and BLR^2 matrices can be compressed by using methods such as SVD, randomized SVD, rank revealing QR, adaptive cross approximation, interpolative decomposition, or learning. Some of these compression schemes do not result in orthogonal bases or diagonal S blocks, but the matrix structure is still called BLR and BLR^2 in previous studies. If everyone renamed the matrix structure every time they changed the compression method, it would be very confusing.
> >
> > Q5. I don't understand the logic of excluding Gaudi-GBLR and BSP+LR from the comparisons, just because they don't have dedicated compression algorithms. It should be trivial to simply use the best compression algorithm for those methods and compare with them.

---

> > > ### Author Response · Authors · 2024-08-14
> > >
> > > Q1: We appreciate the reviewer's feedback regarding potential confusion in naming. However, it's important to note that BLAST matrices are significantly distinct from BLR and BLR^2 matrices regarding the properties of the factored matrices. Moreover, the set of BLAST matrices are not a subset of the set of BLR^2 matrices, vice versa. We believe that using the same name as BLR^2 would lead to greater confusion as these are two distinct matrix structures.
> > >
> > > Q5: Despite our best efforts, we were unable to identify candidate algorithms capable of compressing the dense matrix using BSP+LR or GBLR matrices. This limitation prevented us from including these structures in the data-free compression experiment baselines. We would be extremely grateful if the reviewer could suggest any potential compression methods that we might have overlooked.
> > >
> > > We appreciate Reviewer 2dKz's effort and interest in our work.

---

### Official Review · Reviewer_xrxa · 2024-07-12

**Soundness:** 2
**Presentation:** 3
**Contribution:** 3
**Rating:** 5
**Confidence:** 3

**Summary:**

The authors propose a learnable compressed representation (BLAST) for weight matrices used in deep learning which enables lower complexity matrix multiplications which approximate the full, uncompressed operation.  A BLAST matrix decomposes the original matrix into a grid of blocks of diagonal matrices with shared low-rank factors along each (block) row and column, and these parameters can be learned directly from a random initialization.  Converting a dense matrix to a BLAST matrix by minimizing the Frobenius norm error between original and compressed matrices can be done through gradient descent factorization, and preconditioning can accelerate convergence.  The proposed technique is evaluated in pre-training, fine-tuning, and data-free compression scenarios, using a variety of networks and tasks (vision transformers and LLMs, image classification, image generation through diffusion, and zero-shot language modeling).

**Strengths:**

**Originality**: The proposed approach seems to be entirely novel, building on existing low rank and block sparse techniques.

**Quality**: In general, the major claims are backed up with the necessary experiments - the impact of preconditioning on convergence speed is demonstrated, and the comparisons against the chosen baseline suggest that the BLAST format is superior in many cases, and at least comparable in most.  The breadth of evaluations is welcomed - different types of models and tasks can behave differently to different techniques, but the applicability of BLAST across this variety is demonstrated appropriately.

**Clarity**: The writing is clear, and the organization is fantastic.  A reader can quickly understand what sets BLAST matrices apart from other techniques, if not the importance of its constituent components (see weaknesses below).

**Significance**: Simplifying deep learning in practice, either by reducing memory or compute costs is an important area, and this submission seemingly approaches both angles.  The relative simplicity of the format and the immediate benefits (shown in Table 4) may make this an attractive technique for practitioners to adopt.

**Weaknesses:**

**Originality**: While the authors included a sparse-weight baseline (LLM-Pruner), [SparseGPT](https://arxiv.org/abs/2301.00774) deserves mention (and potentially inclusion as a baseline method to compare against).

**Quality**

The authors claim in line 310 that BLAST matrices allow "more efficient execution in hardware for the same FLOPs (in Figure 5, Figure 6, and Table 1)," but these figures and table don't tell the reader anything about hardware efficiency.  While the authors have presented an algorithm for matrix-vector products, I'm not convinced that matrix-matrix multiplication is as efficient as optimized dense routines tensor execution units in any recent hardware.  (The runtime speedups in Table 4 may simply be due to the memory and bandwidth savings, not an computational gains.)

The authors stress that U and V are shared along rows and columns, but it's unclear why this is so important.  What if each block has its own U and V -- other than increasing the number of parameters, would the factorization process, quality of the results, multiplication algorithms, etc. change?

In line 245, the authors suggest that certain techniques "have not been studied as data-free compression techniques" and are therefore "excluded from the comparison" for this task.  I'm not sure if this is fair or not: were these techniques simply not designed for this task, or are they harder to adapt to this task, or are they entirely incompatible?

**Clarity**: there are some configuration details missing:

- The rank *r* used for the experimental results.  (I see this detail is in the appendix, but I think it's useful to give at least a range of the values used in practice, as was done with *b*, in the main content.)

- The batch size and input sequence length used in the LLM runtime analysis.


Figure 4 is lacking vertical axis labels.

I'm not sure of the difference between, in line 100, "learning" and "finding" the structure of a weight matrix.


**Significance**: the quality of the results lets the submission down, in a way.  Compression rates of 10%-20% are fairly limited, and I'd be curious to know what happens, even if it's catastrophic, at higher compression rates -- even 50% would be informative.  Further, the quality of the BLAST diffusion model's output, while better than SVD's, still fall short of the baseline.  Fine detail is lost (in e.g. the water and rocks of the fourth column, the feathers in the fifth column, and the jersey in the sixth), and larger distortions appear (the hippo's snout no longer looks accurate, the bird's eye has disappeared, and the hot air balloon's shape is extremely lopsided).  Again, these results are better than the other Low-Rank technique, but practitioners may be hesitant to use the technique for this task without significant gains.

**Questions:**

1. Is matrix-matrix multiplication built by performing multiple matrix-vector products (Algorithm 1), or is there something more clever that can be done?

2. What is the difference, in line 100, between "learning" and "finding" the structure of a weight matrix?

3. What is the benefit (other than parameter reduction) in sharing the U and V factors along rows and columns?  Can it be quantified?

4. What rank *r* is used for the experimental results?

5. Regarding skipping some baselines for comparisons, were these techniques simply not designed for this task, or are they harder to adapt to this task, or are they entirely incompatible?

6. What batch size and input sequence length was used for the results in Table 4?

**Limitations:**

The authors' discussion is excellent, only missing one potential further limitation: this technique seems like it'd need extra work to apply to higher-dimensional tensors.  I could imagine applying it on matrix-slices of a tensor, but this may not work well.

---

> ### Author Rebuttal · Authors · 2024-08-06
>
> ### Q1. SparseGPT deserves mention.
>
> We agree that SparseGPT \[Frantar & Alistarh, 2023\] is an interesting related work that deserves mention. SparseGPT uses *unstructured pruning,* which makes it inefficient compared to other structured matrices (including BLAST) for GPU execution since the pruned weights are randomly placed without structured patterns. Also, the SparseGPT compression process is *not* a *data-free* method since it is based on the Hessian matrix for pruning. In contrast, BLAST can be applied to data-free compression as presented in Algorithm 2, and its matrix multiplications with structured sparsity can be accelerated on off-the-shelf CUDA GPUs with significant inference speed up.
> We will include SparseGPU discussion and comparison in the revision.
>
> ### Q2. The result supports improved hardware efficiency but the runtime speedups in table 4 may be due to the memory and bandwidth savings.
>
> Thank you for the insightful feedback\! We agree that memory savings may be the dominant contributor to the runtime improvement in Table 4. We observed that the BLAST matrix multiplication is faster than the dense routine when the weight matrix is large enough. Yet, memory and bandwidth savings are also critical components of hardware efficiency, thus should be rewarded.
> Moreover, there is room for improvement to further optimize the BLAST matrix multiplication kernel using optimized / modified CUDA functions, which we left for future work. The optimized kernel can be realized augmenting the gain from computation savings and the reduced memory access overhead.
>
> ### Q3. The authors stress that U and V are shared along rows and columns, but it's unclear why this is so important.
>
> By sharing U and V factors, we can **significantly reduce the number of parameters** while being able to model multiple structures by the diagonal factors. If the bases are not shared, more parameters are required to achieve a similar level of expressivity.
> Indeed, a BLAST matrix without parameter sharing is identical to a Monarch (i.e., BLR) matrix. Under a similar number of parameters, BLAST matrices outperform BLR matrices in DNNs’ accuracy as we showed in Figures 5, 6, and Table 1.
>
> ### Q5. Why are some baselines skipped for data-free compression?
>
> Please refer to Q2 of “Global Comments.”
>
> ### Q6. Compression rates of 10%-20% are fairly limited. What happens at higher compression rates? . Fine details are missing in the images generated by a diffusion model with BLAST weights, although BLAST shows better image quality than Low Rank.
>
> Compressing the LLMs with structured sparsity (including structured matrix) is extremely challenging. The 20% compression ratio was chosen and accepted in the previous work on structured LLM compression (LLM-Pruner \[30\]). Note that LLM-Pruner is not data-free, whereas BLAST does not need *any* data to compress LLM. Still, BLAST outperforms LLM-Pruner in Table 3 with noticeable gaps.
>
> We conducted additional experiments on 30% compression ratio for diffusion models (Figure A in the attached pdf) and on 40\~50% compression ratio for Llama-7B (Figure B in the attached pdf). Despite the difficulty of the data-free compression task beyond 20% ratio, the additional results show that BLAST preserves more accuracy on zero-shot classification and more details on image generation than the non-learnable baselines.
>
> For the generative tasks  (e.g., Figure 1 and Table 3), we stress that the **goal of model compression is not necessarily to reproduce the outputs of the original model**. The main focus should be on  whether the compressed models’ outputs have details that look realistic (i.e., whether the ‘compressed’ generative model is still a valid model for the generative task). Based on the quantitative results in Table 2, we show that the perceptual quality of the BLAST-generated images is close to the quality of the images generated by the uncompressed model.
>
> ### Q7. Is matrix-matrix multiplication built by performing multiple matrix-vector products (Algorithm 1), or is there something more clever that can be done?
>
> Please refer to Q4 of “Global Comments.”
>
> ### Q8. What is the difference, in line 100, between "learning" and "finding" the structure of a weight matrix?
>
> We differentiate “learning” and “finding” in the following manner: “learning” indicates the structure of a weight matrix is inferred by minimizing the DNN’s prediction loss which involves *datasets*, whereas “finding” is reserved for the *data-free* compression in Algorithm 2\.
>
> ### Q9. What rank r is used for the experimental results?
>
> We vary the rank of the BLAST matrix (from 48 to 192\) to adjust the compression ratio of the models in Figure 5 and Figure 6\. For ImageNet training experiments in Table 1, we use r=128.
>
> ### Q10. What batch size and input sequence length was used for the results in Table 4?
>
> The batch size was set to one, and the length of the input sequence was set to 3\. Specifically, we used the following prompt to let the model generate the desired sequence length: “Increasing sequence: one,” and stopped the generation process when the generated sequence length approaches L=10, 100, or 1000\.
>
> ### Q11. Figure 4 is lacking vertical axis labels.
>
> The normalized Frobenius norm error was used in the vertical axes of Figure 4\. We will include the label.
>
> ### Q12. This technique seems like it'd need extra work to apply to higher-dimensional tensors.
>
> We do not see a straightforward approach to extend BLAST to high-dimensional tensors. We will discuss this as one of the limitations.
>
> We truly appreciate Reviewer xrxa’s feedback. Please let us know whether our comments have successfully resolved all of your concerns or not. If so, we kindly request reconsidering the evaluation of our work.
>
> **Reference**
>
> Frantar E, Alistarh D. Sparsegpt: Massive language models can be accurately pruned in one-shot. In International Conference on Machine Learning 2023 Jul 3 (pp. 10323-10337). PMLR.

---

> > ### Comment · Reviewer_xrxa · 2024-08-08
> > **SparseGPT can target structure, Batched GEMMs are inefficient, image generation results are still unsatisfying**
> >
> > I appreciate the authors' thoughtful responses, but I will keep my rating.
> >
> > Q1. SparseGPT also generalizes easily (as evidenced by the results in the paper and public code) to semi-structured (or N:M) sparsity, which is supported in the hardware and software you used for evaluation (NVIDIA A100 and PyTorch [1]).  They did not experiment on Llama-7B, as you did for the "aggressive compression" results in the new Figure B (thank you for including this!), but for OPT-175B (admittedly, a much larger model), 2:4 sparsity (50% sparsity with a hardware-compatible format) sees only a 1.2 percentage point drop in average zero-shot accuracy, as compared to BLAST's ~15-20 percentage point drop (it's hard to tell precisely what the drop is from the figure) at the same compression rate.  The different model sizes make it hard to compare, but it's at least been shown that SparseGPT can maintain compelling accuracy even with significant compression that is easy for hardware to exploit.
> >
> > Thank you for the reminder that there is a formulation for data-free BLAST compression, which is indeed a difference.  I'd only mention here that SparseGPT uses only 128 calibration samples, and roughly the same amount of computation as BLAST (O(hours) for a single model on a single GPU), so it's otherwise a pretty fair comparison.
> >
> > Further, I'm hesitant to agree that the speedup from BLAST is from reduced FLOPs and not from memory savings, which is also available via unstructured sparsity.  Please read on for more discussion of this point.
> >
> > [1] https://pytorch.org/tutorials/advanced/semi_structured_sparse.html
> >
> > Q2. The efficiency of a batched-GEMM compared to a standard GEMM with the same FLOPs is lower, simply because the per-GEMM arithmetic intensity is lower.  I put together a quick PyTorch script to benchmark a moderately-sized (4Kx4K * 4Kx4K) GEMM, as well as a series of batched GEMMs with increasing batch dimensions, and correspondingly decreasing GEMM-K dimensions, resulting in an overall consistent total FLOP count.  I also added a 2:4 semi-structured sparse GEMM to the mix, since I pointed them out in Q1, above.  Results were measured on a single NVIDIA A100.
> >
> > The latency, in seconds, of 1000 iterations of these workloads are:
> > GEMM: 0.5415s
> > 2:4 GEMM: 0.3463s
> > Batched GEMMs:
> > - b=1: 0.5415s
> > - b=2: 0.5506s
> > - b=4: 0.5723s
> > - b=8: 0.6218s
> > - b=16: 0.9853s
> > - b=32: 1.3376s
> >
> > As you can see, while a small batch dimension is no big difference, as soon as the batch dimension is large enough to start pushing the individual GEMMs into being bandwidth-limited, the overall workload becomes slower.  I also ran a batched gemm with a batch dimension of 16, but with GEMM-K reduced by 32: in this experiment, the total number of FLOPs is halved, but the latency compared to the dense GEMM still increased to 0.6911s.
> >
> > So, while you observe speedups for the workloads you tested, I believe this is due to only the memory reduction, not the FLOPs savings, and custom kernels may not be able to change this behavior.  Unless you can conclusively show that the FLOPs savings are the reason for a speedup in practice, I might consider focusing more on the compression benefits than the reduced FLOPs.  I fully agree that memory and bandwidth savings are critical components of hardware efficiency, but I still disagree that the results presented support the argument that the reduction in FLOPs (as shown in Table 1 and Figure 6) are of major significance.
> >
> > Q3, Q5. Thank you for your responses.
> >
> > Q6. See my point above about an alternate baseline's quality at 50% compression for LLM tasks.  I understand that the goal of BLAST in image diffusion settings is not to match the baseline. Let me clarify my concern: the images generated by BLAST are qualitatively worse than the baseline in a consistent manner.  There is a lack of fine detail (flower, water surface, feathers, jersey), and broad shapes are distorted (hippo, balloon).  If I'm using a model to generate pictures of hot air balloons, I would be unsatisfied with either BLAST or Low-Rank's results in Figure 1, but the baseline model is perfectly acceptable.  I would not judge the compressed model to be useful for the task, but I acknowledge that its results are superior to Low-Rank's.
> >
> > Q7, Q8, Q9. Thank you for your responses.
> >
> > Q10. Thank you for these details, they are helpful to understand the workload under test.  In particular, this is a very short input sequence length, which will reduce the time spent in processing the attention in the model, and will focus the time in the linear layers (which are those that BLAST accelerates).  Further, the relatively larger output sequence lengths will require more processing time in the generative phase, which is extremely bandwidth-limited, as opposed to the context phase, which can be math-limited.  This furthers my suspicion that the speedup you see in practice is entirely unrelated to any reduction of FLOPs.
> >
> >
> > Q11, Q12. Thank you for your responses.

---

> > > ### Author Response · Authors · 2024-08-11
> > > **BLAST GEMM Runtime Evaluation**
> > >
> > > **Q1. SparseGPT can target structure.**
> > >
> > > Thank you for the comments. We completely agree that SparseGPT can be accelerated on the Ampere architecture (or higher version of) NVIDIA GPUs via 2:4 sparsity, although it is our understanding that the 50% compression ratio on LLMs is very challenging to achieve with the structured methods in general (e.g., Structured Pruning \[30\]).
> > > Nonetheless, we would like to clarify that BLAST is less device-specific – one can use BLAST to accelerate the DNN on the legacy NVIDIA GPUs as well as the devices from other vendors. We believe this property will play a crucial role for accessibility of the proposed method.
> > >
> > > Moreover, 2:4 sparsity can be adopted to further compress the BLAST factors or vice versa, which is an interesting combination of two different worlds. We are pleased to work on fusing two methods as future work.
> > >
> > > **Q2. Speedup of BLAST is due to only the memory reduction, not the FLOPs savings, and custom kernels may not be able to change this behavior.**
> > >
> > > Thank you for the insightful feedback and experiments\! We would like to provide additional results on the BLAST GEMM, inspired by your feedback. The results show that **BLAST GEMM can benefit from FLOPs savings.**
> > >
> > > We benchmarked (N\*N x N\*N) dense, 2:4 sparsity, and 50%-compressed BLAST matrix multiplications. The structure of another matrix is fixed to dense. In the table below, we report the cuda time of 1000 iterations using torch.profile on an NVIDIA A100 GPU with 40GB memory.
> > >
> > > To be completely fair with the memory bandwidth, we matched the number of parameters of BLAST to that of 2:4 sparsity, i.e., 17/32 of N^2. In this manner, both 2:4 sparsity and BLAST have the same number of values to be transferred from VRAM to CUDA blocks. In other words, the memory-related latency is similar in both cases. Hence, *if the computation reduction does not contribute to the inference speed-up of BLAST, the runtime of BLAST GEMM will be similar to or slower than the runtime of 2:4 sparsity GEMM.*
> > >
> > > However, BLAST GEMM is **faster** than Semi-structured Matmul when N\>2048 and b=2 (\# blocks=2x2) or N\>4096 and b=4. Based on the fact that their latency related to memory bandwidth is similar, the results indicate that the extra speedup compared to 2:4 sparsity comes from the *FLOPs saving*.
> > >
> > > For reference, we included the runtime of dense GEMM with 50% FLOPs (N\*N x N\*0.5N). The majority of BLAST GEMM runtimes are in between the runtime of 50% dense and 2:4 sparsity, which also shows that the latency improvement of BLAST GEMM is relevant to the FLOPs saving.
> > >
> > >
> > > **Inefficiency of Batched Matmul.** We agree that BMM is less efficient than the dense GEMM of the same FLOPs when the batch size is large. This hinders the efficient CUDA core utilization when the number of blocks is large and each block is small. We will work on circumventing this issue for future work.
> > >
> > > Again, thank you so much for the constructive feedback and discussion. We will discuss this in the revised version. In particular, we will modify Line 296 to “...a larger number of blocks increases the computation and the **memory bandwidth** overhead to perform Equation (5).”
> > >
> > > Table.
> > >
> > > | Models | N=2048 | N=4096 | N=6144 | N=8192 |
> > > | :---- | ----- | ----- | ----- | ----- |
> > > | Dense | 0.339s | 2.556s | 8.198s | 19.325s |
> > > | Dense (50% FLOPs) | 0.209s | 1.313s | 5.098s | 9.861s |
> > > | 2:4 Sparsity | 0.252s | 1.660s | 6.424s | 14.744s |
> > > | BLAST (b=4) | 0.311s | 1.938s | 5.638s | 12.350s |
> > > | BLAST (b=2) | 0.282s | 1.581s | 4.680s | 11.602s |
> > >
> > > **Q10. Very short input length.** We agree that BLAST takes advantage of the short input sequence in Table 4\. However, based on our runtime analysis in Q2, we observed the FLOPs saving is relevant to the runtime reduction.
> > >
> > > **Q6. Image quality degradation.** Thank you for the clarification. We believe there is room to improve the image quality by carefully choosing the layers for compression. But we leave this for future work.

---

> > > > ### Comment · Reviewer_xrxa · 2024-08-12
> > > > **Thank you for the excellent discussion**
> > > >
> > > > **Speedups**
> > > >
> > > > I think this discussion is incredibly useful, and I thank the authors for continuing to explore this area to make sure the benefits are clear.  However, I do not believe their claims have been proven.
> > > >
> > > > Let me take this argument in particular: "if both 2:4 sparsity and BLAST have the same compression rate, then any speedup of BLAST is due to FLOPs savings."  This isn't the case, simply because the relative efficiencies of 2:4 sparse and dense GEMMs are not the same, much less 2:4 sparse and BLAST GEMMs.  Due to the complex nature of parallelizing a GEMM among O(100) processing elements, achieved efficiency (math utilization) can change drastically with small changes in workload size.  For example, while the dense GEMM size in your new table scales roughly linearly with N^3, the same is not true of the 2:4 sparse row; its speedups fluctuate non-monotonically from 1.35x to 1.54x.  From this, I conclude that the speedup of a 2:4 sparse GEMM is a function of more than just FLOPs performed and memory required for the compressed representation: some other factor (or factors) affects the speedup.  The same is clearly true of BLAST matrices: your BLAST4 row's speedups change (monotonically, in this case) from 1.09x to 1.57x.  So, with 2:4's efficiency changing, one can't compare to it directly and suggest that any speedup over *its* results are due to a savings in FLOPs and not the other complicating factors shown to be at play in the 2:4 results.
> > > >
> > > > Similarly, for Q.10, the very short input length means most of the time is spent in the autoregressive generative phase, which uses an effective sequence length of one when generating new tokens.  With a batch size of one (from your original response's Q.10), this makes the effective GEMM size not, for example, 4096x4096 * 4096x4096, but rather 4096x4096 * 4096x1, which will have very different characteristics than the workloads in your most recent table, so it's not reasonable to equate the two.  I used your pseudocode to implement your benchmark and came up with similar results to those shown in your table, but things looked very different for BS=1, seqlen=1.
> > > >
> > > > … all this is to say that understanding speedups is nontrivial, and it is not simply a function of FLOPs and memory.  So, what does that mean for this submission?  I think there are two things to keep in mind:
> > > > 1) provide the readers with some idea of the achievable speedups for the range of BLAST hyperparameters used in practice (b=(2..16), r=chosen to target compression rates of 10% to 75% (I think, based on my understanding of Table 8 in Appendix C)).  This could be as simple as extending your table to include b=8 and 16, or as complete as also providing another similar table with sequence length = 1 (for autoregressive generation) and batch size sweeping from 1 to N (even for a single N).
> > > > 2) avoid attributing some throughput or latency behavior to FLOPs or memory without being very confident in the source of the benefit.  It's safer and less potentially misleading to simply share the throughput and latency behavior in these cases.  Concretely, for Line 296, I might suggest "… a larger number of blocks reduces the efficiency of the batched GEMM operations" - it's accurate, easy to show, and doesn't over-specify the source of the inefficiency.
> > > >
> > > > I am very grateful to the authors for sharing their findings here, and I want to make it clear that I am not weighing my disagreement with their analysis too heavily!  After all, as they say, the speedups should be rewarded, whether they are due to FLOPs or memory savings.
> > > >
> > > > For completeness and inspection, the runnable PyTorch code I used when benchmarking:
> > > > ```
> > > > def blast_matmul(
> > > >         X,  # input,        shape=(B, n, q*b), B=batch_size, n=num_seq
> > > >         U,  # left factor,  shape=(b, p, r), b=num_blocks, r=rank
> > > >         S,  # diag factor,  shape=(b, b, r)
> > > >         Vt, # right factor, shape=(b, r, q)
> > > >     ): # output = X @ A.T where A is a BLAST matrix of shape (b*p, b*q).
> > > >     B = X.shape[0]
> > > >     n = X.shape[1]
> > > >     b = U.shape[0]
> > > >     r = U.shape[2]
> > > >     q = Vt.shape[2]
> > > >     p = U.shape[1]
> > > >
> > > >     X = X.view(b, B*n, q) #rearrange(b, B*n, q)
> > > >     Y = torch.bmm(X, Vt.mT) # multiply right factor
> > > >     Z = Y.unsqueeze(0) * S.unsqueeze(2) # multiply diagonal factor
> > > >     Z = Z.sum(1) # aggregate, shape=(b, B*n, r)
> > > >     Out = torch.bmm(Z, U.mT) # multiply left factor
> > > >     return Out.view(B, n, b*p)
> > > > ```
> > > >
> > > > And my calculation to choose r to hit a target compression rate:
> > > > ```
> > > > r = comp_rate*n / (2 + (b*b/n))  # find the exact r to achieve comp_rate (0.0 .. 1.0) memory savings
> > > > r = int((r//16) * 16)  # set to the next-lowest multiple of 16 for hw-friendliness (next-lowest results in more compression than strictly necessary)
> > > > ```

---

> > > > > ### Author Response · Authors · 2024-08-14
> > > > >
> > > > > Thank you for the insightful and constructive feedback! We acknowledge the reviewer's concern regarding the mechanism behind the speed-ups and appreciate the valuable suggestions. We also believe the source of latency improvement should not be over-specified. We will update the paper accordingly, along with some information on the achievable speed-ups for the hyperparameters. Once again, we appreciate Reviewer xrxa's effort and interest in our work.

---

### Official Review · Reviewer_V5GM · 2024-07-12

**Soundness:** 3
**Presentation:** 3
**Contribution:** 2
**Rating:** 4
**Confidence:** 3

**Summary:**

This paper attempts to find efficient structures in weight matrices of deep learning models. The basic idea is to learn a group of block-wise low-rank matrices via gradient descent. The proposed method replaces original dense weight matrices and hence needs retraining.

**Strengths:**

* The paper introduces an original approach on replacing weight matrices of DNNs to be block-wise low-rank through training.
* The evaluation includes multiple models and different tasks, showing good generalizability of this work.

**Weaknesses:**

* The significance of the proposed method remains unclear. Compared with prior methods such as Gaudi-GBLR [14], the accuracy results of this work do not show consistent superior performance nor with a convincing explanation.
* The method does not seem to improve efficiency much, showing limited significance.

**Questions:**

* How different is this work compared with block sparse + low-rank [23] and Gaudi-GBLR [24]? Why is adaptive and learnable a new contribution when these ideas have already been proposed? The paper [23] is not introduced before use in experimental results.
* In Table 4, when L is 100, the execution time of 20% compression ratio increases compared with no compression, while other two columns show slight reduction of execution time. What is the reason?
* Please provide more details, even if it is obvious, that the learned BLAST matrix structures do not need any library function customization. What is specifically used as the backend compute kernel?

**Limitations:**

Yes.

---

> ### Author Rebuttal · Authors · 2024-08-06
>
> ### Q1. The significance of the proposed method remains unclear. Compared with prior methods such as Gaudi-GBLR \[14\], the accuracy results of this work do not show consistent superior performance nor with a convincing explanation.
>
> Please refer to Q3 of “Global Comments.”
>
> ### Q2. The method does not seem to improve efficiency much, showing limited significance.
>
> Firstly, the number of parameters in the ViT model for ImageNet classification is reduced by 72.2% with BLAST, while achieving **higher** accuracy than Gaudi-GBLR and even the dense baseline. This was done without any hyperparameter tuning. Also, unlike Gaudi-GBLR which requires specialized hardware and/or library to achieve inference speed-up (which has not been demonstrated), BLAST shows significant speed-up on off-the-shelf CUDA GPUs.
>
> Secondly, compressing the LLMs with structured sparsity (including structured matrix) is extremely challenging. The 20% compression ratio was chosen and accepted in the previous work on structured LLM compression (LLM-Pruner \[30\]). Note that LLM-Pruner is not data-free, whereas BLAST does not need *any* data to compress LLM. Still, BLAST outperforms LLM-Pruner in Table 3 with noticeable gaps.
>
> Lastly, we conducted additional experiments on 30% compression ratio for diffusion models (Figure A in the attached pdf) and on 40\~50% compression ratio for Llama-7B (Figure B in the attached pdf). Despite the difficulty of the data-free compression task beyond 20% ratio, the additional results show that BLAST preserves more accuracy on zero-shot classification and more details on image generation than the non-learnable baselines.
>
> For the generative tasks  (e.g., Figure 1 and Table 3), we stress that the **goal of model compression is not necessarily to reproduce the outputs of the original model**. The main focus should be on  whether the compressed models’ outputs have details that look realistic (i.e., whether the ‘compressed’ generative model is still a valid model for the generative task). Based on the quantitative results in Table 2, we show that the perceptual quality of the BLAST-generated images is close to the quality of the images generated by the uncompressed model.
>
> ### Q3. How different is this work compared with block sparse \+ low-rank \[23\] and Gaudi-GBLR \[14\]?
>
> Please refer to Q1 of “Global Comments.”
>
> ### Q4. The paper \[23\] is not introduced before use in experimental results.
>
> Thank you for pointing this out, we will introduce it before the experimental result section.
>
> ### Q5. In Table 4, when L is 100, the execution time of 20% compression ratio increases compared with no compression. What is the reason?
>
> Table 4 actually shows that the execution time **decreases** with compression with L=100. This may have been an error made by the reviewer.
>
> ### Q6. Please provide more details that the learned BLAST matrix structures do not need any library function customization. What is specifically used as the backend compute kernel?
>
> Please refer to Q4 of “Global Comments.”
>
> We truly appreciate Reviewer V5GM’s feedback. Please let us know whether our comments have successfully resolved all of your concerns or not. If so, we kindly request reconsidering the evaluation of our work.

---

> > ### Comment · Reviewer_V5GM · 2024-08-12
> >
> > Thank you for your explanation.

---

### Author Rebuttal · Authors · 2024-08-06

We appreciate all reviewers’ efforts and constructive suggestions. We are encouraged by the reviewers’ positive feedback, specifically in that “the proposed approach seems to be entirely novel” (Reviewer xrxa) and that our approach “shows an improvement in validation accuracy compared to existing methods such as Gaudi-GBLR” (Reviewer 2dKz). In the following, we answer a few concerns raised by multiple reviewers. We also attach a *PDF file summarizing the additional experimental results.*

### Q1. Contributions of BLAST?

Firstly, we clarify the structure of the BLAST matrix to emphasize its differences from existing methods such as LR, BLR, and BSP. The key difference is the flexibility of the BLAST matrix - its unique formulation can **learn the structure** of the weight matrix rather than **forcing it to have a specific (hand-crafted) structure**. For example, as we present in Section 3 and Appendix A.1, based on the diagonal terms $S\_{i, j}$, the BLAST matrix can capture multiple structures. This is done in an efficient manner that consists of equally partitioned blocks with structure-defining parameters, allowing for (i) better accuracy-efficiency tradeoff, (ii) inference speedup, and (iii) effective data-free model compression. BLAST also benefits from low training overhead – BLAST has much less hyperparameters that are less sensitive to training stability compared to those in Gaudi-GBLR and BSP+LR.

The BLAST matrix structure differs from the BLR^2 \[Ashcraft, Buttari & Mary, 2021\] (a related work suggested by Reviewer 2dKz) method in its left, right, and diagonal factors.
(i) The middle factor ($S\_{i,j}$) of a BLAST block ($A\_{i,j}=U\_i S\_{i,j} V\_j^T$) is a diagonal matrix, whereas BLR^2 uses a *low-rank* matrix. Accordingly, a BLAST matrix can capture **high-rank blocks** with **fewer parameters** than BLR^2. (ii) BLR^2 restricts its shared left and right factors to have *orthonormal* bases, which introduces extra overhead and instability at each gradient descent update. In contrast, *none* of the BLAST factors have such orthonormal constraints so that the training process does not change from the conventional DNN training scheme. In revision, we will cite BLR^2 and discuss the work.

### Q2. Exclusion of (some) baselines in experimental results

We excluded BSP+LR (Pixelfly) \[23\] and Gaudi-GBLR \[14\] matrices from the compression benchmark since they do not have dedicated compression algorithms, nor can they be easily decomposed. Comparison with BLR is feasible as  a decomposition algorithm is available \[13\]. Hence, we conducted the data-free compression experiment with the BLR weights on Llama-7B. The BLR format did not outperform the other baselines (or BLAST), as presented in Table A in the attached PDF.

### Q3. Results are not significant, especially compared to Gaudi-GBLR

Firstly, we would like to state that methods such as Gaudi-GBLR (along with others such as BSP+LR) are not designed for **data-free model compression**. This makes our BLAST approach a unique contribution in this context.
Secondly, for pre-training/fine-tuning experiments on CIFAR-10/100   and ImageNet, we had to extensively  tune Gaudi-GBLR’s 4-8 hyperparameters with consultation with the authors of Gaudi-GBLR. Because of this hyperparameter tuning difficulties, Gaudi-GBLR performs worse than our BLAST for the ImageNet benchmark (while the opposite is true for the simpler CIFAR-10/100 benchmark). The difficulty of tuning and training for Gaudi-GBLR makes BLAST weight matrices a stronger contribution.The BLAST weight matrices are much **easier to train** because no hyperparameters are introduced except for the rank and number of blocks. This explains why **BLAST outperforms Gaudi-GBLR on ImageNet** where hyperparameter tuning is more challenging. This indicates that the BLAST matrix is better suited for large-scale training.
Thirdly, BLAST shows the significant inference speedup using an off-the-shelf GPU and standard CUDA library functions, whereas Gaudi-GBLR does not show such inference speedup gains on the same setup as it requires specialized hardware and/or optimized library functions to fully realize its claimed gain.

### Q4. How is BLAST matrix-matrix multiplication done?

The matrix-matrix multiplication between a BLAST matrix and an input X is done with **a few lines of** **basic tensor operations**–batched matrix-matrix multiplication (e.g., torch.bmm), reshaping, transpose, element-wise multiplication, and reduction by sum. We present a brief pseudocode below.

```python
# Pseudocode of Blast Matrix-Matrix Multiplication

def blast_matmul(
        X,  # input,        shape=(B, n, q*b), B=batch_size, n=num_seq
        U,  # left factor,  shape=(b, p, r), b=num_blocks, r=rank
        S,  # diag factor,  shape=(b, b, r)
        Vt, # right factor, shape=(b, r, q)
    ): # output = X @ A.T where A is a BLAST matrix of shape (b*p, b*q).

    X = X.rearrange(b, B*n, q)
    Y = bmm(X, Vt.T) # multiply right factor
    Z = Y.unsqueeze(0) * S.unsqeeze(2) # multiply diagonal factor
    Z = Z.sum(1) # aggregate, shape=(b, B*n, r)
    Out = bmm(Z, U.T) # multiply left factor
    return Out.rearrange(B, n, b*p)
```

**Reference**

Ashcraft C, Buttari A, Mary T. Block Low-Rank Matrices with Shared Bases: Potential and Limitations of the BLR ^2 Format. SIAM Journal on Matrix Analysis and Applications. 2021;42(2):990-1010.

**PDF with additional experiments, Table and Figures**

---

### Decision · Program_Chairs · 2024-09-25

**Decision:**

Accept (poster)

**Comment:**

Overall: this paper describes an interesting variation on low-rank/structured weight matrices with enough experimentation to suggest that the structure might be worth further investigation, and hence sharing with the NeurIPS audience.  The reviewers, particularly xrxa, draw attention to many deficiencies of experimentation and presentation, which it is strongly expected the authors will take into consideration when revising the paper.

On balance, however, I feel that the objections of reviewers V5gm and 2dKz are adequately rebutted, which puts this paper into the "borderline accept" category.  This is largely due to xrxa's observation that the "proposed approach seems to be entirely novel, building on existing low rank and block sparse techniques."  It feels as if it is valuable to have this approach described, even without the custom kernels, and  that would make its impact more widely

As AC, I agree with xrxa: "the speedups should be rewarded, whether they are due to FLOPs or memory savings."
However, this discussion does become relevant when rebutting the observation that a very short sequence length used, which artificially amplifies the performance gains by loading much more work onto the linear layers.  The rebuttal claims that "we observed the FLOPs saving is relevant to the runtime reduction", but this is more contentious given xrxa's observations.